*Report*

# A method for benchmarking genetic screens reveals a predominant mitochondrial bias

Mahfuzur Rahman[1,†] (ID), Maximilian Billmann[1,*,†] (ID), Michael Costanzo[2] (ID), Michael Aregger[2] (ID), Amy H Y Tong[2], Katherine Chan[2], Henry N Ward[3] (ID), Kevin R Brown[2] (ID), Brenda J Andrews[2,4] (ID), Charles Boone[2,4] (ID), Jason Moffat[2,4] (ID) & Chad L Myers[1,3,**] (ID)

## Abstract

We present FLEX (Functional evaluation of experimental perturbations), a pipeline that leverages several functional annotation resources to establish reference standards for benchmarking human genome-wide CRISPR screen data and methods for analyzing them. FLEX provides a quantitative measurement of the functional information captured by a given gene-pair dataset and a means to explore the diversity of functions captured by the input dataset. We apply FLEX to analyze data from the diverse cell line screens generated by the DepMap project. We identify a predominant mitochondria-associated signal within co-essentiality networks derived from these data and explore the basis of this signal. Our analysis and time-resolved CRISPR screens in a single cell line suggest that the variable phenotypes associated with mitochondria genes across cells may reflect screen dynamics and protein stability effects rather than genetic dependencies. We characterize this functional bias and demonstrate its relevance for interpreting differential hits in any CRISPR screening context. More generally, we demonstrate the utility of the FLEX pipeline for performing robust comparative evaluations of CRISPR screens or methods for processing them.

**Keywords** computational evaluation; CRISPR screens; electron transport chain

**Subject Categories** Biotechnology & Synthetic Biology; Computational Biology

**Mol Syst Biol. (2021) 17: e10013**

## Introduction

CRISPR-based screening techniques have become a central instrument for systematic investigation of gene function. At the forefront of such efforts, the Cancer Dependency Map (DepMap) effort aims to catalogue genetic dependencies of all human genes across a range of cultured cell lines spanning various tumor entities. To date, the loss-of-function fitness effects of 17,634 genes have been measured in 563 cell lines (19Q2 data release) (Meyers *et al*, 2017; Dempster *et al*, 2019a). These data provide a comprehensive and easily accessible resource for biological hypothesis generation. Several studies have developed computational methods to systematically derive functional information from these data, including inferring genetic interactions (Rauscher *et al*, 2018) or functional relations by identifying co-essentiality relationships (similarity of genes' dependency profiles) (Boyle *et al*, 2018; Pan *et al*, 2018; Kim *et al*, 2019; preprint: Wainberg *et al*, 2019). Despite the wealth of data and a diversity of methods for processing CRISPR screening data, we lack standard benchmarks for evaluating their ability to extract functional information, which ultimately limits our progress in establishing the best practices for analyzing CRISPR screens.

Here, we developed FLEX (Functional evaluation of experimental perturbations), a pipeline to evaluate functional screening data or algorithms designed to improve scoring or interpretation of such data. FLEX derives reference standards from diverse genome-wide functional resources such as CORUM complexes (Giurgiu *et al*, 2019), curated pathways (Liberzon *et al*, 2011), GO Biological Processes (BP) (Ashburner *et al*, 2000), or genomic data-derived functional networks (Greene *et al*, 2015). It then uses these reference standards to (i) generate summaries of precision-recall (PR) performance on a global and local scale by assessing the degree to which genetic dependency profiles capture known functional relationships, (ii) investigate underlying functional diversity driving the observed PR performance, and (iii) report a diversity-normalized PR statistic that highlights both the quality and functional diversity of functional relationships captured by a dataset of interest. FLEX is available as an R package.

We illustrate the functionality of the FLEX pipeline through several applications on the DepMap collection of CRISPR screens, including comparative benchmarking of alternate versions of the

---
1  Department of Computer Science and Engineering, University of Minnesota – Twin Cities, Minneapolis, MN, USA
2  Donnelly Centre, University of Toronto, Toronto, ON, Canada
3  Bioinformatics and Computational Biology Graduate Program, University of Minnesota – Twin Cities, Minneapolis, MN, USA
4  Department of Molecular Genetics, University of Toronto, Toronto, ON, Canada
   *Corresponding author. Tel: +49 170 7456047; E-mail: maximilian.billmann@gmail.com
   **Corresponding author. Tel: +1 612 624 8306; E-mail: chadm@umn.edu
   †These authors contributed equally to this work

dataset, comparisons of different methods for deriving co-essentiality networks, and an evaluation of the impact of number of screens on the quality of the resulting co-essentiality network. These analyses highlight the prominence of mitochondria-related genes' dependency profiles in CRISPR screens, which we hypothesize is a result of protein stability and screen dynamics.

# Results

### Development of a pipeline for evaluation of CRISPR screen data

We developed FLEX to evaluate the capacity of the DepMap CRISPR knockout co-essentiality networks to recover complex, pathway, and biological process co-membership of human genes (Fig 1A, Appendix Fig S1). Precision-recall (PR) statistics calculated from FLEX showed that co-essentiality scores recapitulated many known functional relationships—for example, at a precision of 0.5, 3,348 true-positive (TP) co-complex pairs from the CORUM complex standard were identified based on pairwise Pearson correlation coefficients derived from the DepMap dataset (Fig 1B). FLEX uses PR statistics to account for the strong class imbalance typically observed in functional genomics data, where the number of positive events (true functional relationships) is much smaller than the number of negative events (unrelated pairs) (Myers *et al*, 2006). While PR statistics provide a general quantification of functional information, they do not provide insight into the diversity of functional information captured by a particular dataset. To understand how individual protein complexes contribute to overall performance, we decomposed the contribution of each complex (number of TP pairs) across the range of precision levels achieved (see Materials and Methods for details). FLEX visualizes these contributions per complex as a "contribution diversity" plot, where at each precision threshold (y-axis), the fraction of TP pairs mapping to each protein complex at that threshold is summarized (x-axis) (Fig 1C).

Precision thresholds dominated by a single color indicate low functional diversity among the gene pairs supporting the predicted functional relationships at that cutoff. As a complementary view of how functional performance varies across functional modules, FLEX also reports the area under the PR curve (AUPRC) for each individual complex along with the complex size (Fig 1D, Table EV1).

Strikingly, we found that only two of 1,697 complexes in the CORUM standard, the electron transport chain (ETC) I holoenzyme and the 55S mitochondrial ribosome, dominate the strongest correlated gene pairs from the DepMap dataset, contributing ~76% of the 3,348 TP pairs at a precision of 0.5 (Fig 1C). Consistent with a predominant functional signal contributed by these complexes, exclusion of the ETC and 55S mitochondrial ribosome annotations from the 1,697-complex standard, but not removal of other large complexes or small complexes with high AUPRC, vastly reduced global PR performance of the DepMap (Fig 1E), suggesting that caution needs to be taken when interpreting such global evaluations. Similar issues have been reported when evaluating other types of genomic datasets in a pairwise manner, particularly for large, coherent protein complexes (Myers *et al*, 2006; Liu *et al*, 2009; Drew *et al*, 2017).

Complexes such as the ETC and the 55S mitochondrial ribosome dominate these global evaluations because they are well-captured by profile similarity in the DepMap data, as supported by focused PR analysis of gene pairs associated with only genes in these complexes (Fig 1C–E, Appendix Figs S2A–D and S3A–D), but due to their large size, they contribute a large number of pairs. To enable functional evaluations of CRISPR screen data that are less influenced by well-performing, large gene sets, we implemented in FLEX an additional, complementary metric, termed module-level Precision-Recall (mPR) performance. To compute the mPR measure, the contribution diversity data (e.g., as reported in Fig 1C) are used to count the number of distinct functional modules in the standard that are represented among the set of gene pairs meeting a given precision threshold (see Materials and Methods for details). These results

**Figure 1. FLEX reveals mitochondrial bias in functional CRISPR/Cas9 screening data.**

A  FLEX inputs a CRISPR screening dataset and functional reference standards to compute gene-level performance and module-level (e.g., protein complex) performance summaries (see Appendix Fig S1 for details).

B  Precision-recall (PR) performance of gene–gene co-essentiality scores using the CORUM complex standard to define true positives (TP). This is a traditional PR curve with the following modifications: (i) the absolute number of TP instead of fractional recall (0-1) on the x-axis (simply a scaling of the axis) and (ii) use of a log-scale on the x-axis (highlights high precision part of the curve). Pearson correlation coefficients (PCC) are computed between CERES score profiles across the 563 19Q2 DepMap screens for all possible gene pairs.

C  Contribution diversity of CORUM complexes to PR performance (B). Functional composition of different complexes (x-axis, as a fraction) to the set of TP pairs predicted at different precision levels (y-axis) are plotted. Only the minimum number of complexes to cover the set of TP pairs (for a certain precision) are considered (see Materials and Methods for details). Complexes with a fraction smaller than 0.01 (1%) at any precision are collectively shown in light gray. The background (bg) contribution diversity represents the functional contribution of complexes across the entire CORUM standard. Highlighted complexes are defined in (D).

D  Size and individual CORUM complex PR performance. Area under the PR curve (AUPRC) was computed per complex on a fractional precision-recall (0-1) scale. Dot size corresponds to the mean within-complex CERES profile PCC, adjusted by the standard error. Protein complexes with at least 30 members (genes) are defined as large, otherwise small. Complexes with an AUPRC of at least 0.4 are defined as high AUPRC, otherwise low. All sub-complexes mapping to the ETC I or 55S mitochondrial ribosome are shown in the respective color.

E  PR performance of gene–gene co-essentiality scores (see (B)). Black line shows complete data, colored lines show the performance after sets of complexes (defined in (C)) were removed from the data and standard. The inset barchart shows the percentage of TP lost at a precision of 0.5 after either set of complexes is excluded.

F  Module PR (mPR) curve summarizes performance at a functional module level (here, CORUM protein complexes). This is a modified version of a precision-recall curve (B) with the number of unique complexes (x-axis) covered and plotted (instead of unique gene pairs) at each precision cutoff (y-axis) (see Materials and Methods for details).

G  Comparison of two methods measuring co-essentiality in the DepMap using PR and mPR plots. The method proposed by Wainberg and colleagues is compared with the standard PCC-based method (top). The well-balanced coverage of complexes is shown after their ETC-related complex exclusion (dotted lines, top) as well as in the mPR curve (bottom). The approach from Wainberg *et al* (2019) bases gene-pair similarity scores on FDR corrected *P*-values (1 - fdr) resulting in a 'late start' of the PR curve (many values at top are the same, 1.0).

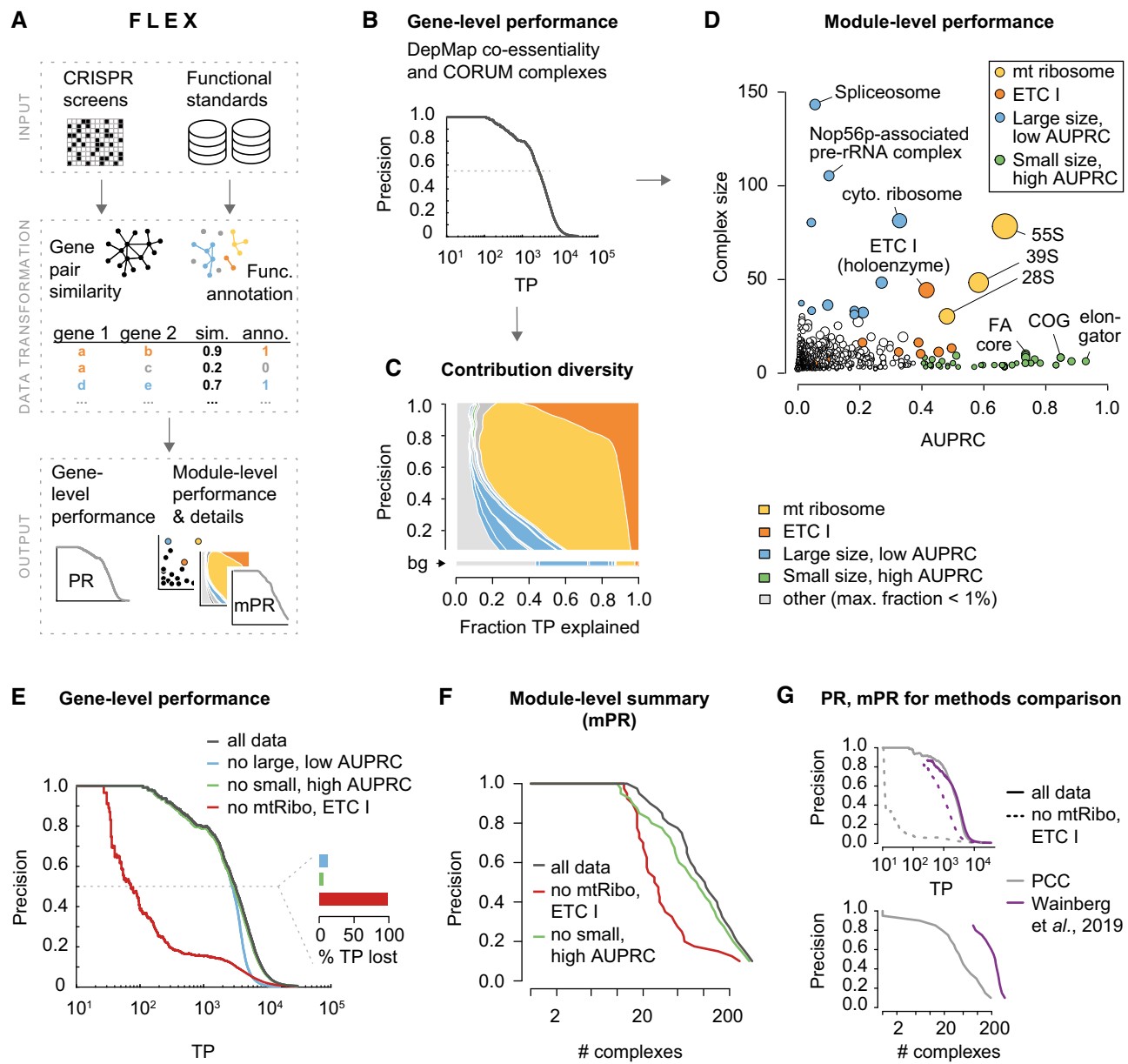

**Figure 1.**

are then summarized across all gene sets, but each gene set is allowed to contribute only a single count to the total displayed on the x-axis, thereby stabilizing the contribution of large versus small gene sets to the evaluation (Fig 1F). We emphasize that each of these complementary FLEX visualizations (Fig 1B, C, D and F) is produced by default for any dataset evaluated by the pipeline; considering all of them collectively is important to gain an accurate perspective of the functional information captured by a given dataset.

### Applications of FLEX to benchmark CRISPR screen data and analysis methods

FLEX enables objective benchmarking of methods for scoring or processing CRISPR screen data, several of which have been recently published specifically for the DepMap (Boyle *et al,* 2018; Pan *et al,* 2018; Kim *et al,* 2019). To demonstrate the utility of FLEX for benchmarking, we provide several example use cases. First, we used FLEX to compare an earlier DepMap data release (18Q3) to a later release (19Q2), which was based on an improved CERES score (preprint: Dempster *et al,* 2019b). The later release shows substantial improvement in capturing functional relationships and a greater functional diversity in the relationships captured (e.g., ETC-related complexes are less dominant) (Appendix Fig S2E–H), suggesting that the improvements to the CERES score have reduced the dominance of the ETC and the 55S mitochondrial ribosome. Second, we used FLEX to benchmark a variety of similarity metrics in their ability to construct co-essentiality networks that capture known functional relationships from the DepMap dataset. Specifically, we evaluated four different similarity measures for

gene pairs: cosine similarity, inner (dot) product, Pearson correlation, and Spearman correlation. We found that Pearson correlation (PCC) and Spearman correlation provide comparable performance and that they clearly outperformed cosine and dot product similarity measures on the DepMap dataset (Fig EV1A and B) (PCC is implemented as the default similarity measure in FLEX). Third, we used FLEX to evaluate a collection of published methods for producing co-essentiality networks from the DepMap data (Boyle *et al*, 2018; Kim *et al*, 2019; preprint: Wainberg *et al*, 2019). We found substantial dependence on the mitochondrial complexes in all of them, with the notable exception of the algorithm published by preprint: Wainberg *et al* (2019), which captures functional relationships with a much greater functional diversity than other methods (Appendix Figs S4 and EV2A and B). This superior performance with respect to functional diversity may result from accounting for covariance among cell lines, which is a key feature of the method developed by Wainberg and colleagues but not others. This difference is clearly highlighted by FLEX's mPR metric (Figs 1G and EV2C and D).

In a fourth application example, we applied FLEX to explore the extent to which the ability to derive co-essentiality networks from a CRISPR screen dataset depends on the number of screens. Specifically, we subsampled different numbers of screens from the DepMap data, measured co-essentiality networks on the resulting datasets of varying size, and evaluated these scores for functional information using FLEX. Our analysis showed that the amount of functional information captured increases with the number of screens included as expected, but that this saturates relatively quickly (Appendix Fig S5). For example, FLEX analysis indicates that there is little measurable difference between the quantity of functional information captured by only 300 screens as compared to the complete collection of 563 in the 2019Q2 release of the DepMap data (Appendix Fig S5). Even a set of as few as 100 randomly sampled screens performs similarly to the complete set of 563. Our FLEX analysis also indicated that with 15 or fewer screens, the ability of co-essentiality scores to accurately capture functional information drops dramatically (Appendix Fig S5), suggesting this is a practical limit on the minimum number of screens required for generating co-essentiality maps.

As a final example FLEX application, we explored the question of how the identity of genetic screens affects the type of functional information captured in co-essentiality scores. Specifically, we applied FLEX to analyze the co-essentiality scores derived from 31 genome-wide CRISPR-Cas9 screens against 27 DNA-damaging agents (Olivieri *et al*, 2020). Interestingly, FLEX contribution diversity analysis showed a strong dominance of protein complexes related to DNA damage repair (e.g., Fanconi anemia complex, DNA ligase IV−XRCC4−XLF complex, DNA synthesome complex) among predicted functional relationships (Appendix Fig S6A and B). At the same time, ETC-related complexes were not strongly represented among these co-essentiality scores, suggesting that the factors driving the variation in ETC-related genes' phenotypes are less prominent in this context. This example more generally shows how the biological focus of the investigated set of screens, an experimental theme spanning various model organisms (Jonikas *et al*, 2009; Billmann *et al*, 2018), can be evaluated.

## Exploring the basis of dominant ETC-related co-essentiality relationships in CRISPR screens

Given the dominance of the functional signal contributed by ETC-related complexes in DepMap co-essentiality relationships, we further explored the basis of this observation. First, we compared dependency data from 149 cell lines in the 19Q2 DepMap that had been screened both at the Broad Institute (hereafter referred to as Broad DepMap) and the Sanger Institute (Sanger DepMap). Since we also observed a strong signal for the ETC V complex (Appendix Fig S3A), we hereafter consider ETC I, V, and the 55S mt ribosome and refer to them collectively as ETC-related complexes. While the dependency profiles for the same cell lines generally agree across these two datasets (Dempster *et al*, 2019a), we found the ETC-related genes to be among the protein complexes exhibiting the strongest differences between them, with the Broad DepMap consistently measuring stronger dropout phenotypes for these genes (Fig 2 A). Assay length is a major difference between Broad and Sanger DepMap screening protocols (Broad screens are conducted over 21 days while the Sanger screens are completed over 14 days) (preprint: Dempster *et al*, 2019b), and thus, we reasoned that the difference in ETC-related genes' phenotype observed in the Broad and Sanger screens may be related to screen sampling times. Specifically, we hypothesized that the rate at which functional proteins are cleared from the cell after successful gene disruption may impact phenotypic penetrance over the course of a screening experiment. In other words, the growth phenotype associated with disruption of an essential gene would only be observed after the corresponding essential protein is mostly depleted from the cell population. Thus, in the case of a highly stable essential protein, cells may need to be cultured for a longer period of time after gene disruption to observe the resulting growth defects. Consistent with this hypothesis, we found that protein complexes with significantly more severe fitness defects in the Broad DepMap screens (FDR < 5%) tend to be more stable ($z$-score > 1; $P = 0.001$, hypergeometric test), based on analysis of available protein half-life data derived from monocytes, B cells, and hepatocytes (Mathieson *et al*, 2018). Strikingly, the ETC I and V complexes showed the highest protein stability of any complex in the CORUM standard (Fig 2B, Appendix Fig S7A–C). In contrast, the 55S mitochondrial ribosome had a protein half-life comparable to the median complex half-life. The phenotypic delay observed for the 55S ribosome may also be related to high remaining protein levels of the ETC, which acts downstream of the 55S ribosome. More specifically, the phenotypic effect of the 55S ribosome perturbation is a result of its impact on ETC complex disruption (i.e., the ETC complex is epistatic to the 55S ribosome in this context), which may explain why it exhibits similar dynamics in the context of a CRISPR screen.

To test whether temporal drop-out patterns that are dependent on assay length and whether protein stability could contribute to the high similarity of ETC-related co-essentiality profiles, we performed several CRISPR/Cas9 screens in a single cell line (HAP1 cells) and measured gene essentiality at multiple different time points over the course of the screen (Fig 2C). Specifically, we sampled cells every 3–4 days following initial infection and compared the abundance of guide RNAs (gRNAs) targeting a particular gene at a given time point relative to the starting gRNA abundance for the corresponding gene. Applying this approach, we generated dynamic essentiality

profiles, derived from seven biological replicate screens, for each of ~18,000 genes targeted by our genome-wide TKOv3 gRNA library (Fig 2C, Appendix Fig S8A–C). Similar to observations from the Broad DepMap, we found that genes with similar time-resolved essentiality profiles, derived from a single HAP1 cell line, tended to be functionally related, and often annotated to the same protein complex or biological pathway (Fig 2D–F). Strikingly, using FLEX to dissect the observed functional performance revealed that the same ETC-related complexes were responsible for the majority of the functional associations derived from our dataset (Fig 2D–G, Appendix Fig S9A–D). We note that other large essential protein complexes with shorter protein half-lives (e.g., the 26S proteasome or the cytosolic ribosome) drop out relatively rapidly when targeted (within 5 days post-puromycin selection), while the ETC-related complex members take substantially longer, dropping out between 6 and 13 days post-puromycin selection (Figs 2H and EV3A–C). This observation is consistent with phenotypes for these complexes observed in earlier CRISPR/Cas9 screens (Tzelepis *et al*, 2016) and RNAi-based screens (Marcotte *et al*, 2012). In our own FLEX-based analysis of RNAi screens (McFarland *et al*, 2018), we observed a similar, albeit weaker, enrichment for mitochondrial ribosome-related gene pairs (Appendix Fig S10), although unlike CRISPR

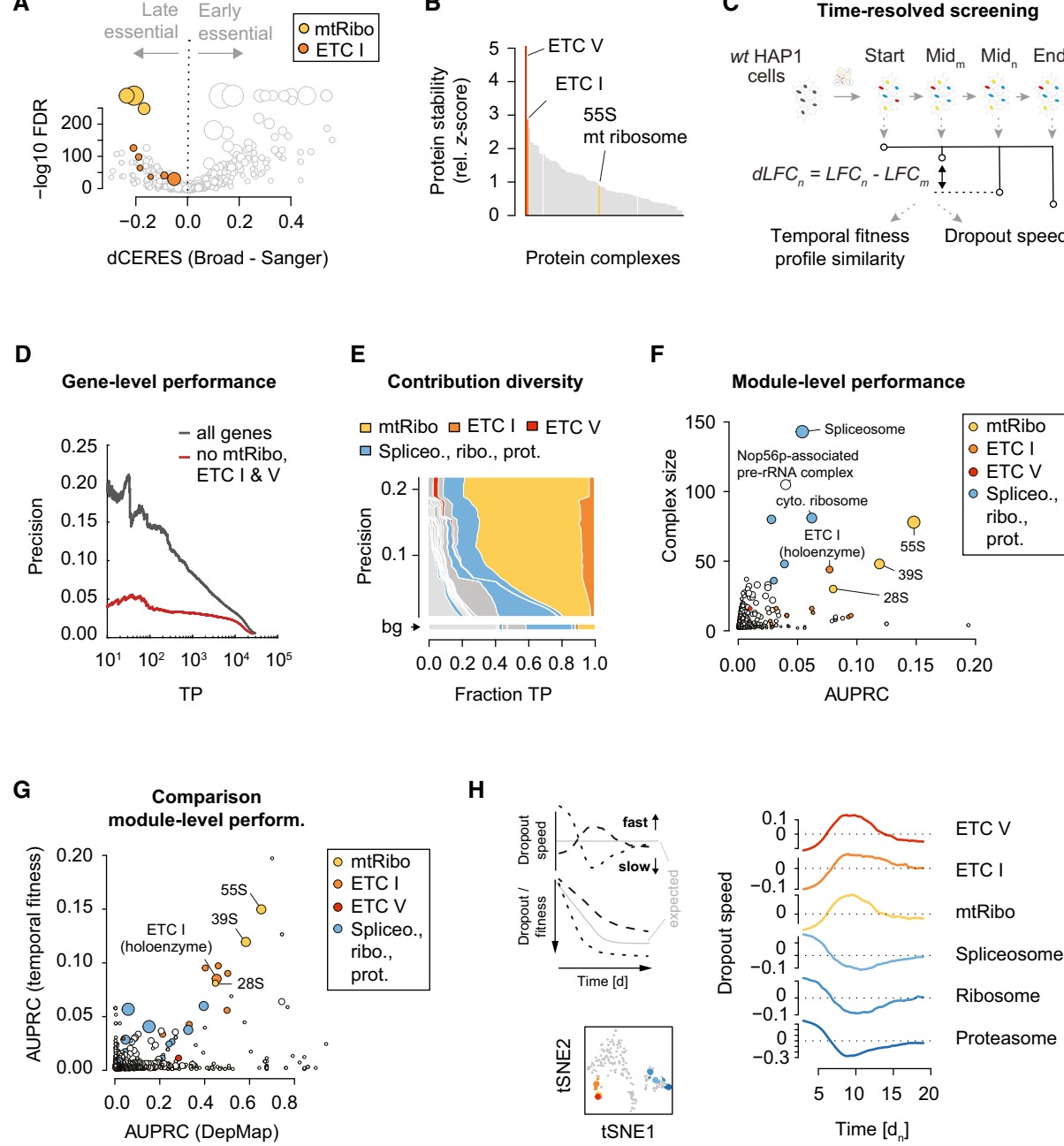

**Figure 2.**

**Figure 2.   Delayed ETC CRISPR/Cas9 fitness phenotypes create within-complex co-essentiality.**

A   Protein complex-level differences in fitness effects between the Broad and Sanger DepMap screens. The 149 cell lines and 16,464 genes common to both datasets are compared. For each CORUM complex, the median differential CERES score (x-axis) and a paired Wilcoxon rank sum P-value with BH-correction are shown. Mitochondrial ribosome (yellow) and ETC I (orange) sub-complexes are highlighted. Dot size is proportional to complex size.

B   Protein stability of CORUM complexes. Protein half-life data were taken from B cells, hepatocytes, and monocytes, and summarized on the CORUM complex level. Half-life data were z-transformed, and the minimum z-score set to 0 to emphasize large z-scores. Complexes for which at least 5 members contributed data across the three cell lines are shown.

C   Scheme of time-resolved genome-wide CRISPR/Cas9 screens in HAP1 cells. Temporal fitness profile similarity was estimated by computing the pairwise PCC between genes with 32 unique measurements across time. The dropout speed was derived from profiles interpolated from the 32 measurements after correcting for maximal dropout effects (see Materials and Methods).

D   Precision-recall (PR) curve showing HAP1 temporal fitness profile similarity performance using CORUM complexes as a pairwise functional standard. Black line shows complete data, red line performance after ETC I, V, and mitochondrial ribosome (ETC-related complexes) are removed from the data and standard.

E   Contribution diversity of HAP1 temporal fitness profile similarity PR performance using the CORUM complex standard. Shown are the fraction of TP pairs for CORUM complexes (distributions across the x-axis) at different precision cutoffs (down the y-axis). The minimum number of complexes to cover the complete set of TPs is shown (see Materials and Methods). Complexes with a fraction smaller than 0.01 (1%) at any precision are collectively shown in light gray. The background (bg) functional diversity represents the distribution of categories across the entire reference standard (i.e., the expected distribution in a random selection of gene pairs).

F   Module-level performance of HAP1 temporal fitness profile similarity shows CORUM complex size and AUPRC. Dot size corresponds to the mean within-complex similarity, adjusted by the standard error. All sub-complexes mapping to the ETC-related complexes are shown in the respective color.

G   Comparison of module-level performance between Broad DepMap co-essentiality and temporal fitness. AUPRC measures the performance of each dataset in reconstructing CORUM complex co-memberships. Dot size is proportional to complex size.

H   Dropout speed for ETC-related and other selected essential complexes. Dropout speed is a normalized estimate of the derivative of an LFC profile (across time) for each guide (see Materials and Methods). A positive dropout speed indicates faster relative dropout, while a negative dropout speed indicates slower dropout (see left panel for hypothetical LFC profile examples and their corresponding dropout speeds). The average dropout speed across all genes in each of the indicated complexes is plotted as a function of screen sampling time (right). tSNE embedding groups CORUM complexes with similar dropout speed (see Materials and Methods). The six selected complexes on the right are indicated in the tSNE plot (large colored dots) and sub-complexes are labeled with matching colors (bottom).

screens, co-essentiality scores from RNAi screens also exhibited dominant enrichment for cytoplasmic ribosome gene pairs (Appendix Fig S10).

Given the delayed phenotype of protein complexes with long protein half-lives, we reasoned that the time at which a CRISPR/Cas9 screening experiment was sampled could affect the measured dependency on a particular gene target, and vice versa, that the measured dependency may reflect differences in effective sampling time. To test this, we first sorted the 563 cell lines in the Broad DepMap using the median CERES score for ETC-related complexes. As expected, while genome-wide CERES scores for each cell line exhibited comparable ranges (Fig 3A, gray), ETC-related CERES scores strongly varied across cell lines (Fig 3A, red, Appendix Fig S11A–C). Furthermore, when we added the data from the 149 Sanger DepMap screens that overlapped the Broad DepMap, a matched comparison of the ranks for those 149 showed lower ranks (weaker ETC signal) relative to the corresponding Broad screens (Fig 3B). We further tested how different time points of a single cell line, HAP1, would rank within the Broad DepMap collection of screens based on the strength of the ETC-related phenotype. We leveraged 27 time points measured in 7 independent genome-wide screens taken between 6 and 19 days post-puromycin selection (Appendix Fig S8A–C) (see Materials and Methods). We found that those HAP1 screen time points spanned the range of Broad DepMap screen ranks, with early time points showing weaker dependency on ETC-related genes (Fig 3C). Notably, the strength of the ETC-related dependency itself predicted HAP1 screen timepoints with reasonable accuracy ($r = 0.61$, $P = 0.0007$). This was not true of the dependency scores for other essential complexes such as the 26S proteasome ($r = -0.22$, $P = 0.28$), the spliceosome ($r = -0.01$, $P = 0.96$), or the cytosolic ribosome ($r = 0.37$, $P = 0.055$) (Fig EV3D–F). Together, this suggests that the strength of ETC-related fitness phenotypes is able to accurately recover the effective length of time a screen was cultured before gRNA abundance was quantified.

# Discussion

Our analysis suggests a link between the strength of ETC-related gene dependency and the screen sampling time. In the context of an effort like the DepMap, which is focused on screening large collections of diverse cell lines, there may be a complex interplay between global protein stability, screen sampling time, and doubling rate of the cell line being screened (Fig 3D). While there is likely true variation in the extent of genetic dependency on mitochondria function across different cell types and genetic backgrounds, we speculate that a substantial portion of the quantitative differences observed in the strength of the ETC-related phenotypes in the DepMap may instead reflect differences in the effective sampling time, cell line doubling rate, and protein stability across these cell lines. While this effect is readily discoverable in the DepMap dataset, phenotypes for these ETC-related genes should be interpreted with caution in other CRISPR screen contexts as well, especially if one is interested in scoring differential phenotypes (e.g., cell line-specific dependencies, genetic– or chemical–genetic interactions).

Why are ETC-related genes unique in this regard? If differences in the effective sampling timing or growth rates of cell lines are limited to ± ~50% of the typical sampling time in a given collection of screens, one would expect that only protein complexes like the mitochondrial ribosome and ETC I genes, whose fitness effect size is still increasing even late in screens would show different phenotypes related to such differences in timing. Other essential complexes drop out rapidly enough that there is negligible variation in phenotypes for the vast majority of screens regardless of small variation in effective sampling time or other factors. We note that while multiple lines of evidence support our hypothesis about the

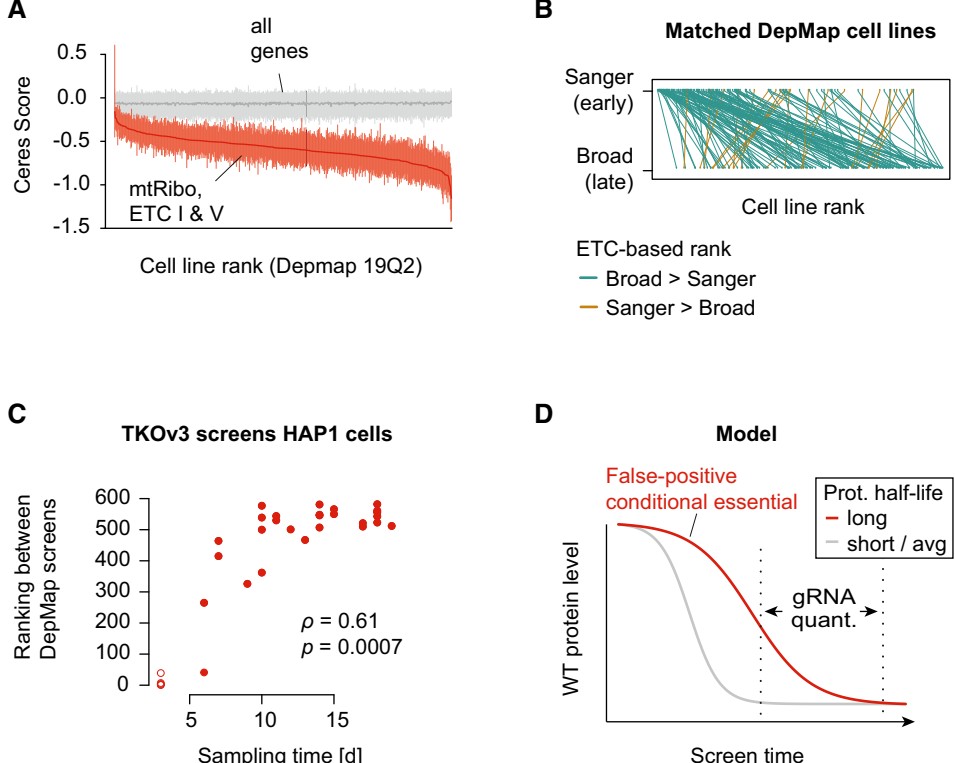

**Figure 3. CRISPR screen sampling time and protein level change.**

A Broad DepMap genome-wide CRISPR/Cas9 screens ranked by the median CERES score across the ETC-related complexes. The middle red line indicates the median, the vertical lines the 25 and 75% quantiles of a given screen. Gray lines represent the same metrics for all genes in the genome.

B Pairwise comparison of Broad and Sanger DepMap screens based on their median CERES score of ETC-related complexes. Highlighted are 149 cell lines common to both datasets. To rank those cell lines, Sanger data from those 149 screens were added to the 563 Broad DepMap screens and all screens were ranked. Green lines indicate a higher ranking of the Broad screen (assay length 21 days) and brown a higher ranking for the corresponding Sanger screen (assay length 14 days).

C Rank of HAP1 time course genome-wide screens in the Broad DepMap screens based on the adjusted median ETC-related LFC. HAP1 screens were performed with the TKOv3 library, and LFC values were adjusted by centering non-essential genes around 0 and core essential genes around −1 (see Materials and Methods). HAP1 screens sampled at T3 are shown as circles indicating that they have not been used for computing the Spearman's rank correlation coefficient and the associated statistical significance (see Materials and Methods for details).

D Wild-type protein abundance of two protein complexes is schematically displayed over the course of a CRISPR screen. The measured phenotype (e.g., gRNA abundance as a proxy for cell fitness) depends on the presence of a sufficient amount of protein to fulfill a cellular function. Stability of proteins, the rate of cell doublings that redistribute residual protein, protein levels required for normal function, or more stable epistatic protein complexes determine the penetrance of cellular fitness phenotypes throughout the course of a CRISPR experiment.

effect of protein stability on ETC-related genes' phenotypes in CRISPR screens, more definitive experiments could be done to further test this hypothesis. For example, one could specifically quantify the dynamics of wild-type protein abundance in a population of cells expressing guides targeting ETC-related genes. Also, we note that beyond sampling time, growth rate, and protein stability there are likely additional non-genetic factors, such as the redox potential of the media as explored by Lagziel and colleagues (Lagziel *et al*, 2019), that could similarly modulate the apparent phenotypes measured in CRISPR screens.

Our study also highlights the utility of the FLEX pipeline, which enables objective benchmarking of functional relationships and informative summaries of the underlying functional diversity. The focus of our example applications of FLEX described here is evaluation of co-essentiality scores derived from (single knockout) genome-wide CRISPR screens. However, we emphasize that the

FLEX pipeline can more generally evaluate the quality/functional composition of pairwise gene relationships of any type. For example, FLEX could be used to directly evaluate genetic interactions derived from combinatorial (double targeting) screens (Gonatopoulos-Pournatzis *et al*, 2020), protein–protein interactions (Luck *et al*, 2020), co-expression relationships, or the output of machine learning approaches focused on inferring related gene pairs. Also, FLEX is flexible in the sense that the functional standard used as the basis for evaluation can be readily changed. Users can easily choose between any of the four standards included (e.g., see Appendix Fig S2), or an external standard can be loaded as long as it defines a relationship between two genes. Importantly, we note that the utility of FLEX is inherently limited by the reference standard(s) used. Functional annotations for the human genome are incomplete and nonuniform in their coverage, and these biases can influence the relative performance of datasets evaluated by FLEX. A

good practice to achieve robust conclusions is to evaluate the datasets of interest against multiple different functional standards using the FLEX pipeline. In general, resources such as FLEX and objective applications of them to existing data and processing methods are critical to our effective interpretation of large-scale CRISPR screens and other functional genomic data.

# Materials and Methods

## Reagents and Tools table

| Reagent/Resource | Reference or source | Identifier or catalog number |
|---|---|---|
| **Experimental models** | | |
| HAP1 cells (*H. sapiens*) | Horizon Genomics | CVCL_Y019 |
| HEK293T cells (*H. sapiens*) | ATCC | CRL-3216 |
| **Recombinant DNA** | | |
| lentiCRISPRv2 | Addgene | Cat #52961 |
| pMD2.G (envelope plasmid) | Addgene | Cat #12259 |
| psPAX2 (packaging plasmid) | Addgene | Cat #12260 |
| **Oligonucleotides and other sequence-based reagents** | | |
| *TKOv3 gRNA library* | Addgene | Cat #90294 |
| **Chemicals, enzymes, and other reagents** | | |
| Wizard Genomic DNA Purification Kit | Promega | Cat #A1120 |
| DMEM low glucose | Wisent | Cat #319-162-CL |
| DMEM high glucose | Life Technologies | Cat # 11995-065 |
| Opti-MEM | Life Technologies | Cat #31985-070 |
| X-tremeGene 9 DNA transfection reagent | Roche | Cat #06365809001 |
| Plasmid maxi purification kit | Qiagen | Cat #12963 |
| Fetal Bovine Serum (FBC) | GIBCO | Cat #12483-020 |
| Puromycin | Wisent | Cat #400-160-UG |
| **Software** | | |
| R version 3.6.3 | https://www.r-project.org/ | N/A |
| org. Hs.eg.db_3.10.0 | https://bioconductor.org/packages/release/data/annotation/html/org. Hs.eg.db.html | N/A |
| Bowtie v0.12.8 | http://bowtie-bio.sourceforge.net/index.shtml | N/A |
| **Other** | | |
| Illumina HiSeq2500 | Illumina | |

### Methods and Protocols

FLEX is designed to perform a systematic functional evaluation of genome-scale perturbation data. It has three different components: generation of reference standards, gene-level (global) evaluation, and module-level (local) evaluation. To use FLEX, the user must provide an input dataset and select a reference standard to evaluate against. FLEX enables both global and local functional evaluations and supports a number of visualization options (Appendix Fig S1).

### *Generation of reference standards*
To systematically evaluate functional relationships between gene pairs, FLEX uses various public reference datasets. A majority of these datasets include genes grouped into different modules (a set of related genes); for example, in the CORUM reference dataset, a module refers to a protein complex. Relationships between all possible gene pairs from all modules form a co-annotation (co-membership) based binary reference standard. In such a reference standard, gene pairs co-annotated to the same module (within-module pairs) are labeled positives (1) and gene pairs from two different modules (between-module pairs) are labeled negatives (0). For all the positive pairs, the source(s) of their co-annotation (module IDs) are stored.

A single reference standard provides an exclusive view of biological complexity. To support functional evaluation from multiple perspectives, FLEX supports four different reference standards. For protein complexes, FLEX uses CORUM v3.0 (Giurgiu *et al*, 2019) as the reference standard. For pathways, it uses MSigDB (Liberzon

*et al,* 2011) that collates several pathway datasets, and for GO (Ashburner *et al,* 2000), it uses biological processes (BP). For reference standards based on complexes and pathways, a positive example is defined as a gene pair annotated to the same complex (or pathway). A gene pair forms a negative example when the genes come from two different complexes (or pathways). In contrast, for the GO BP dataset, FLEX first applies a filter based on the number of genes annotated to a GO term (term size). Biological processes that are too specific (term size < 10) or too general (term size >= 300) are excluded. Then using the gene annotations in the filtered GO BP candidates, FLEX applies a similar approach as outlined for complexes and pathways to define the GO BP-specific reference standard.

While all of these aforementioned reference standards are manually curated and thus high quality, they lack in terms of their genome-wide coverage. For a broader standard, FLEX includes an integrated, data-driven reference functional network named GIANT (Greene *et al,* 2015), which reports inferred functional relationships from many different genomic or proteomic data sources. A node represents a gene in this network, and an edge represents an inferred functional relationship between two genes. While CORUM, Pathway, and GO BP provide annotations for 3,662, 8,904, and 13,637 genes, respectively, GIANT covers ~25K genes. To transform the GIANT network to a reference standard, the gene–gene relationships (edges) in the network are first ranked by the relationship strength (edge weights), in descending order. Next, the top one million gene–gene relationships are labeled as positives (the rest are negatives), resulting in a density of ~2.6% for the positive standard. The density of positives for the CORUM, Pathway, and GO BP standards are ~0.6, ~7, and ~6%, respectively.

Even though FLEX provides four different reference standards by default, users can easily define additional reference standards as desired. The only requirement is that the new reference standard provides associations between a set of gene pairs. Therefore, any dataset with modules or any network quantifying gene–gene relationships is appropriate as input to FLEX to generate reference standards.

Analysis performed using FLEX can be divided into two broad categories (details below): gene-level (global) evaluation and module-level (local) evaluation (contribution diversity, module-level performance, and module-level summary). Module-level analyses are only feasible when the reference standard has a modular hierarchy such that genes are grouped into modules (e.g., CORUM complexes, pathways, etc.). Reference standards without a module-level hierarchy (e.g., GIANT) are only limited to gene-level performance analysis.

### Gene-level evaluation

FLEX performs gene-level evaluations using genome-wide quantitative perturbation effects. These effects can either be dependency scores (Meyers *et al,* 2017), where each gene in the library is systematically knocked out across a panel of cell lines, or genetic interaction (GI) scores (Aregger *et al,* 2020), where a single gene is first knocked out and a panel of gene knockouts are introduced by library screening in the mutant background. Then, depending on user input, FLEX either calculates a pairwise profile similarity score for each gene pair (using gene profiles across screens/experiments) or uses the direct measurements between gene pairs. As FLEX evaluates gene pairs, direct measurements are only relevant when the screens also represent genes (a second knockout for GI, for example). Profile similarity scores between gene pairs are meaningful in

either case (GI or dependency). FLEX uses Pearson correlation coefficient (PCC) values as measures of profile similarity.

Once a pairwise measurement for pairs of genes and their corresponding co-annotations from a reference standard are available, FLEX calculates how well the measurements agree with the co-annotations. A traditional way to capture this agreement is to use a receiver operating characteristic (ROC) curve. However, as all of our reference standards are highly imbalanced (i.e., positive to negative ratio is small), a more appropriate metric to use is precision-recall (Myers *et al,* 2006; Saito & Rehmsmeier, 2015). FLEX uses a precision-recall (PR) curve to summarize the gene-level (global) functional performance, although it plots the number of TPs (equivalent to recall on a log scale) on the x-axis instead of recall.

$$Precision = \frac{TP}{TP + FP}$$

$$Recall = \frac{TP}{TP + FN}$$

### Contribution diversity

As biological annotation standards are modularized, FLEX computes a contribution diversity matrix to understand how different modules contribute to the overall performance. Due to inherent redundancies among modules, a true positive in the standard is sometimes associated with multiple modules. To account for redundancies at the module level, FLEX estimates a subset of modules that explains all of the TPs for a set of precision levels from the gene-level PR curve. Calculating such a subset optimally is an NP-hard problem (Binshtok *et al,* 2007), and hence, FLEX uses a greedy approximation algorithm (shown below). To illustrate the algorithm (and all methods from here on), we will use CORUM as the reference standard and CORUM complexes as modules.

Algorithm: calculate *contribution diversity*.
Input: a set of precision thresholds, P, profile similarity PCCs, CORUM standard.
Output: a 2D matrix, C (row: CORUM complexes, col: precision thresholds, entry: number of TP pairs uniquely contributed by the complex at that precision threshold).
Calculate sets of TPs (also the associated complexes) for all precision thresholds, P
for i = 1 : |P| (length of P)
 Q <- set of TPs at P[i]
 while Q is not empty
 Compute the number of TPs associated with individual complexes.
 Rank the complexes by the number of TPs contributed (descending).
 S <- TPs associated with the highest ranked complex, j
 C[j,i] = |S| (number of contributions for complex j)
 Q <- SetDifference(Q,S) (remove S from Q)
 end while
end for

The algorithm outputs a precision versus complex contribution matrix that is next visualized using a Muller plot. This is termed the contribution diversity plot, and it visualizes the diversity of complexes that constitute global performance. When applied to the

CORUM reference standard, the contribution diversity analysis reduces the number of effective complexes to 1,697 (out of 2,916 total complexes), highlighting the minimal set of required complexes to explain the functional performance.

### Module-level performance

A module-level performance evaluation encapsulates the local performances of individual modules (e.g., complexes). For each complex in the CORUM reference standard, FLEX generates a per-complex subset of the reference standard that includes gene pairs between all of the genes from the complex (within-complex pairs) and pairs between genes from the complex and genes from the rest of the complexes (between-complex pairs). For each complex, within-complex pairs constitute the positive standard whereas between-complex pairs comprise the negative standard. Using these per-complex standards, FLEX calculates an area under the PR curve (AUPRC) for all individual complexes. This is visualized using a scatter plot with the AUPRC values on the x-axis and the size of the complexes on the y-axis.

### Module-level summary

FLEX generates a module-level summary plot to account for the disproportionate nature of modules in genome-wide datasets. It first calculates a module-level precision-recall (mPR) metric by computing a contribution diversity matrix (contribution diversity, method) and then counting, at each precision, the number of modules that are represented.

FLEX then visualizes this using a module-level summary plot that outputs the number of represented modules along the x-axis and precision values along the y-axis. To qualify, a complex must contribute at least one TP pair toward the global performance and 10% of all of the possible within complex (TP) pairs must be present at that precision.

### Pooled CRISPR HAP1 dropout screens

Pooled CRISPR dropout screens, including CRISPR library virus production and virus titer determination, were performed as described recently (Chan et al, 2019; Aregger et al, 2020). In brief, human HAP1 cells were obtained from Horizon Discovery (wt: clone C631, sex: male with lost Y chromosome, RRID: CVCL_Y019) and maintained in DMEM, low glucose (10mM), 1mM glutamine, 10% FBS.

CRISPR library virus production was performed in HEK293T cells. Therefore, 10 million cells were seeded per 15-cm plate in DMEM medium containing high glucose, pyruvate, and 10% FBS. Twenty-four hours after seeding, the cells were transfected with a mix of 8 µg lentiviral lentiCRISPRv2 vector containing the TKOv3 gRNA library (Hart et al, 2015) (Addgene #90294), 4.8 µg packaging vector psPAX2, 3.2 µg envelope vector pMD2.G, and 48 µl X-tremeGene 9 transfection reagent (Roche) in 1.4 ml Opti-MEM media (Life Technologies) for a total volume of 800 µl. Virus-containing media was harvested 48 h post-transfection.

For pooled CRISPR dropout screens, 3 million HAP1 cells were seeded in 15-cm plates in 20 ml of specified media. A total of 50–-90 million cells were transduced with the lentiviral TKOv3 library at a MOI~0.3, so that each gRNA is represented in about 200–300 cells. 24 h post-infection, transduced cells were selected in 1 µg/ml puromycin for 48 h. Cells were then harvested and pooled, and 30 million cells were collected for subsequent genomic DNA

extraction and determination of the library representation at day 0 (i.e., T0 reference). The pooled cells were seeded into three technical replicate plates, each containing 15 million cells (> 200-fold library coverage) and passaged every 3–4 days and at > 200-fold library coverage until T18. Cell pellets from each replicate were collected at each timepoint of cell passage.

Genomic DNA was extracted using the Wizard Genomic DNA Purification Kit (Promega). Sequencing libraries were prepared from 50 µg of the extracted genomic DNA in two PCR steps, the first to enrich guide-RNA regions from the genome, and the second to amplify guide-RNA and attach Illumina TruSeq adapters with i5 and i7 indices. Barcoded libraries were gel purified, and final concentrations were estimated by quantitative RT–PCR. Sequencing libraries were sequenced on an Illumina HiSeq2500 instrument using single-read sequencing. The T0 and T18 time point samples were sequenced at 400- and 200-fold library coverage, respectively.

### Mapping of reads to gRNAs

FASTQ files from single-read sequencing runs were first trimmed by locating constant sequence anchors and extracting the 20 bp gRNA sequence preceding the anchor sequence. Trimmed reads were aligned to the TKOv3 library reference using Bowtie (v0.12.8) allowing up to 2 mismatches and 1 exact alignment (specific parameters: -v2 -m1 -p4 --sam-nohead). Successfully aligned reads were counted and merged along with annotations into a matrix.

### LFC precision-recall analysis

To control quality of genome-wide CRISPR/Cas9 screens in HAP1 cells, gene-level fitness effects were estimated by first computing a log2 fold-change (LFC) quantifying the dropout of a gRNA from the population between T0 (after puromycin selection) and a given mid or end point (T3 - T19). The LFC values of the four gRNAs targeting a given gene were mean summarized. Gold-standard essential (reference) and non-essential (background) gene sets were taken from Hart et al, (2015) and Hart et al (2017). The identification of reference (essential) genes using LFC values of a given screen was assessed by computing precision-recall statistics.

### Calculation of ranks for HAP1 screen time points

Using the common essential and non-essential gene sets that were used for scaling the DepMap CERES scores, we scaled the gene dropout effects in HAP1 cells. At each time point between T3 and T19, LFC values, which represent the difference between a given time point and T0, were scaled to ensure a median score of −1.0 for the essential genes and 0 for the non-essential genes. We then merged each of the 31 HAP1 screen time points with the 563 Broad DepMap screens and calculated median scores for a subset of genes (genes from ETC-related complexes, spliceosome, 26S proteasome, cytoplasmic ribosome). Finally, we ranked all HAP1 screens (time points between T3 and T19) separately by the calculated median score. Spearman's rank correlation between the resulting ranks for all 27 screens sampled between T6 and T19, and the time points at which a given screen had been sampled was computed using the R function 'cor.test' and the statistical significance was based on 25 degrees of freedom. Note that due to strongly varying drop out patterns of core essential genes as well as strongly variable LFC values for non-essential genes at very early time points (T3), LFC

scaling generated several extreme values. Therefore, to compute more conservative correlations coefficients between sampling time and ranking, the four T3 screens were excluded.

### Estimating the gene and protein complex dropout speed

For each of the 71k gRNAs in the TKOv3 library, 31 LFC measurements were taken between T3 and T19 in wild-type HAP1 cells. A loess model was fit through the 31 measurements and T0, estimating the interpolated LFC at 0.2 days resolution. All possible differential (d)LFC values were computed by contrasting interpolating LFC values 3 days apart:

$$dLFCn = LFCn - LFCm;$$

where $n$ is between T3 and T19, m is between T0 and T16, and $n - m$ is equal to 3 days. Furthermore, LFC is the interpolated LFC value for a given gRNA. Since absolute dLFC values depend on the maximal dropout (Fig EV3D and E), the time point of the maximal dropout was estimated by maximizing the separation of non-essential and core essential gene LFC values. The gRNA-level dependency of dLFC values on the LFC at this maximum dropout point was removed by computing the residuals from a Loess fit (Fig EV3D and E). For each gene, gRNA residuals are mean summarized at each point between T3 and T19 to define the dropout speed. For each of the CORUM complexes, the respective gene-level dropout speed was median summarized.

### Code availability

The FLEX R package can be obtained from https://github.com/csbio/FLEX_R.

## Data availability

Processed LFC data from all time-resolved genome-wide CRISPR screens in HAP1 cells are provided in Table EV2.

**Expanded View** for this article is available online.

### Acknowledgements

We thank members of the Myers, Moffat, Boone, and Andrews laboratory for fruitful discussions. This research was funded by grants from the National Science Foundation (MCB 1818293), the National Institutes of Health (R01HG005084, R01HG005853), the Canadian Institutes for Health Research (MOP-142375), Ontario Research Fund, Genome Canada (Bioinformatics and Computational Biology program), and the Canada Research Chairs Program. M.B. was supported by a DFG Fellowship (Bi 2086/1-1).

### Author contributions

Study conception: MB, MR, and CLM; Software and analysis: MR and MB; Result interpretation: MR, MB, MC, HNW, KRB, CB, JM, and CLM; Experiments: AHYT, KC, and MA; Manuscript drafting: MC, MA, HNW, KRB, BJA, CB, JM. MB, MR, and CLM; Funding: BJA, CB, JM, and CLM.

### Conflict of interest

J.M. is a shareholder in Northern Biologics and Pionyr Immunotherapeutics, and is an advisor and shareholder of Century Therapeutics and Aelian Biotechnology. The authors declare no conflict of interest.

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
