## [Review Process File · Molecular Systems Biology]

A method for benchmarking genetic screens reveals a predominant mitochondrial bias

Rahman Mahfuzur, Maximilian Billmann, Michael Costanzo, Michael Aregger, Amy Tong, Katherine Chan, Henry Ward, Kevin Brown, Brenda J. Andrews, Charles Boone, Jason Moffat, and Chad Myers
DOI: 10.15252/msb.202010013

Corresponding author(s): Maximilian Billmann (maximilian.billmann@gmail.com) , Chad Myers (cmyers@cs.umn.edu)

Review Timeline:

Submission Date:	22nd Sep 20
Editorial Decision:	22nd Oct 20
Revision Received:	16th Feb 21
Editorial Decision:	19th Mar 21
Revision Received:	19th Apr 21
Accepted:	20th Apr 21

Editor: Jingyi Hou

Transaction Report:

Thank you for submitting your work to Molecular Systems Biology. We have now heard back from the three reviewers who agreed to evaluate your manuscript. As you will see from the reports below, the reviewers acknowledge the potential interest of the study. They raise however a series of concerns, which we would ask you to address in a major revision.

Since the reviewers' recommendations are rather clear, there is no need to reiterate all the points listed below. Some of the key issues that would need to be addressed are the following:

- A comparison of the presented approach to existing related methods needs to be performed, as Reviewer #2 suggested.
- Reviewer #3 is concerned about the utility and relevance of the presented method in a broader biological context, which needs to be carefully addressed.
- In line with Reviewer #2's comment regarding the Dempsey et al, attention should be given to placing the findings in the context of existing literature and to highlighting the novelty of the current study.

All other issues raised by the reviewers need to be satisfactorily addressed as well. As you may already know, our editorial policy allows in principle a single round of major revision and it is therefore essential to provide responses to the reviewers' comments that are as complete as possible.

On a more editorial level, we would ask you to address the following issues.

REFeree REPORTS

Reviewer #1:

Rahman et al. proposed FLEX, a pipeline for benchmarking genome-wide CRISPR screen data. Using this tool, they identified strong correlation of fitness among ETC-I genes. Further time-resolved CRISPR screens suggest this correlation reflect screen dynamics and protein stability effects rather than genetic dependency.

I enjoyed reading this concise manuscript. FLEX can be a useful tool in the functional screening analysis. The observation of ETC-I related "bias" and the possible explanation is interesting, which should be known by many people who are using large-scale genome-wide CRISPR screen datasets for their research. Meanwhile, I have several concerns about the description of FLEX, as well as the interpretation of the observations related to ETC-I modules.

Major points:

1. It is my understanding that FLEX is applicable for the studies involves gene-gene paired relationship, such as co-dependency maps or genetic interactions. Is it applicable for screening datasets that involves only a few samples (<10)? What is the minimum number of cell lines in the application? The scope of application and the limitation of FLEX should be clearly described in the manuscript.
2. The authors provided evidence which suggests the correlations of ETC-I genes are due to the protein stability and screen dynamics. If this is true, I'd expect the same correlations to be observed in high-throughput shRNA screens in the DepMap data. The authors should check the co-dependency of the ETC genes in shRNA datasets.
3. Many proteins other than the ETC-I genes are highly stable. Do these proteins show co-dependency with the ETC-I genes? Are these genes correlated with ETC genes in the time-course screening experiment?

Minor points:

1. What is the rationale to use different scaling scheme in the x-axis of the PR plots in Fig. 1b&e (exponential), Fig. 1c (fraction of TP), and Fig. 1f&g (# of complexes)? This will impact the calculation of AUC score.
2. Figure 1c is difficult to understand. What are the meanings of colored areas in the plot? It is my understanding that, the authors want to show mt. ribo. module and ETC-I module significantly contribute to the predictive power, whereas some functional modules are not highly predictable. This concept can be visualized in a simpler way to improve the readability.
3. In Fig. 1g, bottom panel, the curve of Wainberg et al does not start from x=0, probably due to visualization issue.

Reviewer #2:

Rahman, Billmann and colleagues present a method (FLEX) for assessing the ability of co-essentiality, derived from CRISPR screens, to predict co-complex / co-pathway membership. Typically one might benchmark such approaches using a single list of true positives (e.g. all gene pairs that belong to the same complex) and a list of true negatives (e.g. all gene pairs that belong to different complexes). The authors show that such an aggregated evaluation set can mask

important signals in the data - in particular for CRISPR screens they show that a couple of mitochondrial complexes are responsible for many of the true positives correctly identified by co-essentiality approaches. They develop, and advocate for, an approach to summarise the performance of co-essentiality across all individual modules. They also provide nice visualisations showing the contribution of each module to the predictive power of co-essentiality methods. Much of the manuscript is devoted to an analysis of which complexes are identified as essential in CRISPR screens performed by the Broad but not the Sanger. The authors provide a nice explanation (long protein half-lives and differing CRISPR screen lengths) for why some complexes are more essential in the Broad screens, but this analysis is largely orthogonal to the establishment of FLEX as a useful benchmark.

Overall FLEX seems like it might be a useful way of benchmarking approaches to identify functional interactions from co-essentiality, but there is very limited benchmarking performed in the paper. The analysis of which modules are essential in the Broad but not Sanger screens is interesting and the newly performed time-course CRISPR screens in HAP1s support the authors' hypothesis that screen duration is the major factor. However, previous work has discussed the issue of the consequences of the different assay lengths between the two institutes and shown an enrichment of mitochondrial genes identified as uniquely essential in the Broad study (Dempsey et al, Nature Comms 2019 <https://doi.org/10.1038/s41467-019-13805-y>). It's not clear that the FLEX analysis adds to this discussion.

Major points:

It is well established that methods to predict functional associations between genes can be biased by certain complexes. Liu et al (NAR 2008, <https://doi.org/10.1093/nar/gkn972>) show that signals from using co-expression to predict complex membership are driven by a small number of large complexes (ribosome, proteasome). Often authors have developed ad-hoc solutions to address issues with large complexes dominating the signal e.g. Drew et al (MSB 2016 <https://doi.org/10.15252/msb.20167490>) exclude complexes with more than 30 subunits from their analysis. This does not diminish the utility of FLEX, but these existing approaches merit some discussion.

Given that the stated purpose of the tool is benchmarking, very little benchmarking is actually performed in the manuscript. Different groups have established different pipelines for scoring CRISPR screens (e.g. the Broad's DepMap project uses CERES, while the Sanger's project uses BAGEL and CRISPRcleanR) but these are not benchmarked. Furthermore, there are many ways of assessing co-essentiality (e.g. Pearson's correlation, Spearman's correlation, cosine similarity) and these are not benchmarked.

Minor points:

Figure 2h - the fast slow arrows are extremely confusing. I initially thought they indicated that ETC/mtRibo were fast drop outs while spliceosome/proteasome were slow drop outs. The differing y-axes also make the chart harder to read.

Figure S4 - the different x-axes used for the module-level summaries make these charts very hard to compare. I thought PCC was performing at a similar level to Kim et al, although this is not the case

Reviewer #3:

FLEX is a pipeline that is used to integrate, among others, publicly available reference datasets to evaluate the integrity of large-scale loss-of-function screening experiments. By using precision recall statistics, CRISPR screening dataset can be interrogated for known functional relationships between gene co-dependency gene pairs and pathways. The authors used FLEX to assess gene-level precision recall performance on commonly published datasets. Analyzing DepMap data based on a CORUM standard, they found that the genes part of the ETC I, V and the 55S mt ribosome complexes are largely dominating the co-essentiality dependency networks. To account for the large effect of certain pathways in determining gene-level PR, the authors developed modular PR to normalize highly co-dependent gene sets into one score. To explain this difference in ETC dependency, the authors hypothesized that the screen length between Broad and Sanger DepMap protocols are different, suggesting that the difference in screen sampling times could contribute to the differences in essentiality. Notably, they have pointed out that the ETC I and V have the highest protein stability, while the 55S was comparatively close to the median half-life. To test this observation, the authors employed a CRISPR/Cas9 screen in HAP1 cells and sampled every 3-4 days. The authors identified several early and late dropout pathways and genes, highlighting the dependency of ETC-related genes in early time points. The authors suggested that the strength of ETC-related fitness is able to accurately predict the time of the screen. At the current stage, it requires a major revision. Specific comments are:

1. Overall, the manuscript presents a method to use a novel AUPRC method to identify differences in co-essential genes between studies. This represents a technical advance to anyone who wants to perform essentiality CRISPR-KO or other KO screens. However, it remains hypothetical that the authors claiming the differences in dropout rate for the screens could be explained by the protein half-life. In particular, the strength of ETC-related phenotype is proposed as a metric for determining the true length of a screen before sampling. While this point could be greatly beneficial, it is uncertain whether this can be true for broad biological contexts. It is also unclear how useful this metric would be compared to any of the other essential genes or pathways. While the technical challenge has been clearly described, the reviewer is not convinced of how this finding would translate to promoting a clear biological advance.
2. The manuscript is written in a blended way, combining both a computational method with some related discoveries/hypotheses. It is confusing as the title only states one of the technical challenges in CRISPR screening analysis. There is a lack of clarification of challenges in the introduction and their approach to address these challenges. There is also a lack of summary on how their method and their discoveries can change the field. Much efforts are needed to make the whole manuscript more consistent, clearer, and less confusing.
3. The reviewer would like the authors to comment more on how monitoring ETC-related phenotypes can be applied to identify the difference in timing for screens between different cell types, and how tracking the dropout rate of ETC-related essential genes are different from doing so with other essential genes or complexes. Do the cells applied with different drugs or other treatments have similar essentiality scores for these genes? The reviewer would like the FLEX analysis of a published screen with cell and/or animal models with drug treatments such as those found in chemotherapy studies to compare the AUPRC. Aside from protein degradation, as suggested by looking at the protein half-life in Figure 2b, there could be other biological reasons for functional differences in dropout speed including different levels of essentiality to the cell. For example, a gene that is required for mismatch repair may drop out much slower than a gene required for cell division checkpoints. Could the proteins of ETC genes be more tolerant of DNA mismatches? In an effort to isolate protein stability from the other biological effects, the reviewer suggests the authors to provide protein stability-corrected essentiality scores for the genes based on the 18,000 gene sgRNA library applied to the HAP1 cells.
4. In Figure 2h, the authors highlight the differences in protein degradation speeds as they relate to dropout rate. The reviewer would like the authors to comment towards if there is a delayed fast or a

delayed slow rate, and if 20 days is enough to show these curves. The reviewers would also like the authors to comment on why the differences in essential/non-essential gene separation in S6C seem to dissipate after 10 days while the ETC complex displays a dropout difference between the Sanger and Broad screens which are between 14 and 21 days?

In conclusion, the reviewer believes that the paper should be considered after more direct demonstration of how the technical bias can be addressed or used to benefit future screens.

Response to reviewers

We would like to thank all the reviewers for their constructive feedback and suggestions. We have addressed all comments, and we agree these changes have improved the manuscript considerably. Below we present a point-by-point response to the reviewer's comments. Reviewer comments are colored in black, our responses are colored in blue, and any changes to the manuscript are colored in red.

Reviewer #1

Rahman et al. proposed FLEX, a pipeline for benchmarking genome-wide CRISPR screen data. Using this tool, they identified strong correlation of fitness among ETC-I genes. Further time-resolved CRISPR screens suggest this correlation reflect screen dynamics and protein stability effects rather than genetic dependency.

I enjoyed reading this concise manuscript. FLEX can be a useful tool in the functional screening analysis. The observation of ETC-I related "bias" and the possible explanation is interesting, which should be known by many people who are using large-scale genome-wide CRISPR screen datasets for their research. Meanwhile, I have several concerns about the description of FLEX, as well as the interpretation of the observations related to ETC-I modules.

Major Points

1. It is my understanding that FLEX is applicable for the studies that involve gene-gene paired relationships, such as co-dependency maps or genetic interactions. Is it applicable for screening datasets that involve only a few samples (<10)? What is the minimum number of cell lines in the application? The scope of application and the limitation of FLEX should be clearly described in the manuscript.

We thank the reviewer for this interesting question. In general, FLEX can be used to evaluate the quality/functional composition of pairwise gene relationships of any type. The reviewer raises an interesting question about the extent to which the ability to derive co-dependency scores from a screening dataset depends on the number of samples. FLEX can be used to provide a definitive answer to this question. To show this, we performed an analysis in which we subsampled different numbers of screens from the Broad DepMap data. Our analysis showed that the amount of functional information captured increases with the number of screens included as expected, but that this saturates relatively quickly (Appendix Figure S5). For example, FLEX analysis indicates that there is little measurable difference between the quantity of functional information captured by only 300 screens as compared to the complete collection of 563 in the 2019Q2 release of the DepMap data (Appendix Figure S5). Even a set of as few as 100 randomly sampled screens performs similarly to the complete set of 563. Our FLEX analysis also indicated that with 15 or fewer screens, the ability of co-dependency scores to accurately capture functional information drops dramatically (Appendix Figure S5), suggesting this is a practical limit on the minimum number of screens required for generating co-dependency maps.

We also explored the related question of how the identity of the screens affects the type of functional information captured in response to Reviewer #3's question. Briefly, in that context, we applied FLEX to analyze the data from 31 DNA-damage related chemical genetics screens from Olivieri *et al*, 2020. Interestingly, FLEX unveils enrichment for many DNA damage response related complexes (Appendix Figure S6), suggesting the type of screens conducted (in this case, phenotypes resulting from exposure to DNA-damage agents) influences the composition of the functional information captured. We believe that these additional analyses provide an informative demonstration of the utility of FLEX, and we thank the reviewer for the interesting questions.

To better demonstrate the scope of application and limitations of FLEX, we added two new paragraphs at the end of subsection 'Example applications of FLEX to benchmark CRISPR screen data and analysis methods':

"In a fourth application example, we applied FLEX to explore the extent to which the ability to derive co-essentiality networks from a CRISPR screen dataset depends on the number of screens. Specifically, we subsampled different numbers of screens from the DepMap data, measured co-essentiality networks on the resulting datasets of varying size, and evaluated these scores for functional information using FLEX. Our analysis showed that the amount of functional information captured increases with the number of screens included as expected, but that this saturates relatively quickly (Appendix Figure S5). For example, FLEX analysis indicates that there is little measurable difference between the quantity of functional information captured by only 300 screens as compared to the complete collection of 563 in the 2019Q2 release of the DepMap data (Appendix Figure S5). Even a set of as few as 100 randomly sampled screens performs similarly to the complete set of 563. Our FLEX analysis also indicated that with 15 or fewer screens, the ability of co-essentiality scores to accurately capture functional information drops dramatically (Appendix Figure S5), suggesting this is a practical limit on the minimum number of screens required for generating co-essentiality maps.

As a final example FLEX application, we explored the question of how the identity of genetic screens affects the type of functional information captured in co-essentiality scores. Specifically, we applied FLEX to analyze the co-essentiality scores derived from 31 genome-wide CRISPR-Cas9 screens against 27 DNA-damaging agents (Olivieri *et al*). Interestingly, FLEX contribution diversity analysis showed a strong dominance of protein complexes related to DNA damage repair (e.g. Fanconi anemia complex, DNA ligase IV–XRCC4–XLF complex, DNA synthesome complex) among predicted functional relationships (Appendix Figure S6A, B). At the same time, ETC-related complexes were not strongly represented amongst these co-essentiality scores, suggesting that the factors driving the variation in ETC-related genes' phenotypes are less prominent in this context. This example more generally shows how the biological focus of the investigated set of screens, an experimental theme spanning various model organisms (Billmann *et al*; Jonikas *et al*), can be evaluated."

2. *The authors provided evidence that suggests the correlations of ETC-I genes are due to the protein stability and screen dynamics. If this is true, I'd expect the same correlations to be*

observed in high-throughput shRNA screens in the DepMap data. The authors should check the co-dependency of the ETC genes in shRNA datasets.

Based on the reviewer's suggestion, we performed a FLEX analysis of a large RNAi screen dataset that includes more than 700 diverse cancer cell lines (McFarland *et al*, 2018). In general, we found that co-essentiality scores based on this RNAi dataset performed substantially less well in capturing known functional relationships than the DepMap CRISPR screen dataset (563 screens) (Appendix Figure S10), which is likely due to the increased targeting efficacy and precision of CRISPR as has been reported in previous studies (Hart *et al*, 2015). Regarding the reviewer's specific question, we did find evidence for enrichment for the 55S mitochondrial ribosome-related genes (Appendix Figure S10B, C) amongst the top-most correlated pairs (they explain ~35% of the true positive pairs at a precision of 50%, Appendix Figure S10C). However, the extent to which these pairs dominate the co-essentiality network here is lower than for any CRISPR screen dataset we have analyzed. We also note that the co-essentiality network from these RNAi screens is composed of a large number of pairs from the cytoplasmic ribosome (Appendix Figure S10C), which is less true for analysis based on the CRISPR screen dataset.

We have added the following text to reflect the results in the second paragraph of the subsection 'Exploring the basis of strong ETC-related co-essentiality relationships in CRISPR screens': "In our own FLEX-based analysis of RNAi screens (McFarland *et al*), we observed a similar, albeit weaker, enrichment for mitochondrial ribosome-related gene pairs (Appendix Figure S10), although unlike CRISPR screens, co-essentiality scores from RNAi screens also exhibited dominant enrichment for cytoplasmic ribosome gene pairs (Appendix Figure S10)."

3. Many proteins other than the ETC-I genes are highly stable. Do these proteins show co-dependency with the ETC-I genes? Are these genes correlated with ETC genes in the time-course screening experiment?

We thank the reviewer for this interesting suggestion. We tested the hypothesis whether genes with high protein stability show fitness profile similarity in the DepMap CRISPR screens and our HAP1 time course screen data. In the DepMap data, we tested this hypothesis by first filtering down to a set of genes where we could expect this trend.

First, about 8000 genes had good quality protein stability in at least one of the three cell lines considered. We further focused on the subset of these genes that showed a detectable fitness effect in a large number of cell lines (>30%; as reference, ETC-I components show a fitness effect in on average 40% of cell lines). The remaining genes that did not belong to the 55S ribosome or the ETC I or V complex (ETC-related genes are labeled as orange in the scatter plots below)

were defined as stable (half-life > 120 h; n = 551) and unstable (half-life < 120 h; n = 102). Neither group showed strong average similarity to ETC I or V genes (see barchart).

A similar trend was observed for more lenient thresholding of genes based on their fitness effect (effect in ≥ 10 of 563 screens):

We performed a similar test on our HAP1 time course screen data. Since all genes with fitness defects tend to show strong correlation with each other, we first normalized the time course profiles by removing the first singular vector from the data after applying SVD (this SV contributed 44% of the variance).

This reduces non-specific correlations between genes with fitness defects. After this correction, more stable proteins showed a weakly elevated temporal profile correlation with ETC I and V complex members (see figure below). However, these differences were not statistically significant (p-val = 0.33 for ETCI, p-val = 0.32 for ETCV; Wilcoxon rank-sum test).

Minor Points

1. What is the rationale to use different scaling scheme in the x-axis of the PR plots in Fig. 1b&e (exponential), Fig. 1c (fraction of TP), and Fig. 1f&g (# of complexes)? This will impact the calculation of AUC score.

We apologize for the confusion. The goal and the input data for those three different types of figures are different. Figure 1B and 1E are traditional precision-recall curves in gene-pair space

with the only modifications being that (1) we plot the absolute number of true positives (TP) instead of fractional recall (simply a scaling of the x-axis) and (2) to emphasize differences in the high-precision part of the curve, we use a log-scale. Figure 1C is designed to pair with the corresponding precision-recall curve (e.g. Figure 1C directly complements Figure 1B), and it plots the functional composition of different functional modules (in this case protein complexes) (x-axis) to the set of true positives predicted at each precision level (y-axis). For a given threshold on precision (horizontal line), the fraction of true positive (TP) pairs contributed by each indicated functional module is plotted, such that the total contribution always sums up to 1. For example, at a precision threshold of 1, the ETCI module contributes more than 50% of the TP pairs. Finally, Figure 1F and 1G (bottom panel) present a modified version of a precision-recall curve that summarizes performance at a functional module level. These curves are assessing precision of gene-pair relationships identified by co-dependency scores, but the recall axis is measured in terms of the number of unique functional modules covered (in this case, protein complexes) rather than unique gene pairs.

We only calculate AUPRC (the area under the curve for curves similar to Figure 1B) values in FLEX, and these are performed on gene-pairs, but using a usual precision (0-1 scale) and fractional recall (0-1 scale) to ensure that the area under the curve never exceeds 1. Hence, the AUPRC values are not affected by the exponential scaling of the x-axis (which is for visualization purposes only).

We understand that two of these types of plots (Fig. 1C, Fig. 1F/G) are non-standard forms of plots that require extra attention for readers to interpret. However, we argue that these are the key utility of the FLEX method-- precision-recall analysis of pairwise co-dependency scores (or other predictions of functional relationships) should indicate the functional diversity of the pairs captured or important information is lost. Our analysis demonstrates that co-dependency scores tend to be dominated by fairly specific functional modules, which is why we are suggesting that users benchmarking data or methods should consider these complementary means of visualizing comparative results.

To avoid this confusion, we revised the corresponding legends and extended the main text to make these points more clear. We clarified legend of figure 1B as: “**B**, Precision-recall (PR) performance of gene-gene co-essentiality profile correlation using the CORUM complex standard to define true positives (TP). This is a traditional PR curve with the following modifications: (1) the absolute number of TP instead of fractional recall (0-1) on the x-axis (simply a scaling of the axis) and (2) use of a log-scale on the x-axis (highlights high precision part of the curve). Pearson correlation coefficients (PCC) are computed between CERES score profiles across the 563 19Q2 DepMap screens for all possible gene pairs.”

For Figure 1C, we modified the legend as: “**C**, Contribution diversity of CORUM complexes in a PR performance (B). Functional composition of different complexes (x-axis, as a fraction) to the set of TP pairs predicted at different precision levels (y-axis) are plotted. Only the minimum number of complexes to cover the set of TP pairs (for a certain precision) are considered (see Methods for details). Complexes with a fraction smaller than 0.01 (1%) at any precision are collectively shown in light grey. The background (bg) contribution diversity represents the functional contribution of complexes across the entire CORUM standard. Highlighted complexes are defined in D.”

We clarified the AUPRC calculation with the revised legend: “**D**, Size and individual CORUM complex PR performance. Area under the PR curve (AUPRC) was computed per complex on a fractional precision-recall (0-1) scale. Dot size corresponds to the mean within-complex CERES profile PCC, adjusted by the standard error. Protein complexes with at least 30 members (genes) are defined as large, otherwise small. Complexes with an AUPRC of at least 0.4 are defined as high AUPRC, otherwise low. All sub-complexes mapping to the ETC I or 55S mitochondrial ribosome are shown in the respective color.”

Legend for Figure 1F is modified to: “**F**, Module PR (mPR) curve summarizes performance at a functional module level (here, CORUM protein complexes). This is a modified version of a precision-recall curve (B) with the number of unique complexes (x-axis) covered and plotted (instead of unique gene pairs) at each precision cutoff (y-axis) (see Methods for details).”

Additional text to explain 1C better is included in the first paragraph of subsection ‘Development of a pipeline for evaluation of CRISPR screen data’: “To understand how individual protein complexes contribute to overall performance, we decomposed the contribution of each complex (number of TP pairs) across the range of precision levels achieved (see Methods for details). FLEX visualizes these contributions per complex as a “contribution diversity” plot, where at each precision threshold (y-axis), the fraction of TP pairs mapping to each protein complex at that threshold is summarized (x-axis) (Figure 1C). Precision thresholds dominated by a single color indicate low functional diversity among the gene pairs supporting the predicted functional relationships at that cutoff. As a complementary view of how functional performance varies across functional modules, FLEX also reports the area under the PR curve (AUPRC) for each individual complex along with the complex size (Figure 1D, Table EV1).”

We also expanded the text to clarify 1F and its connection with 1C in the third paragraph of subsection ‘Development of a pipeline for evaluation of CRISPR screen data’: “To compute the mPR measure, the contribution diversity data (e.g. as reported in Figure 1C) is used to count the number of distinct functional modules in the standard that are represented amongst the set of gene pairs meeting a given precision threshold (see Methods for details).”

2. Figure 1c is difficult to understand. What are the meanings of colored areas in the plot? It is my understanding that, the authors want to show mt. ribo. module and ETC-I module significantly contribute to the predictive power, whereas some functional modules are not highly predictable. This concept can be visualized in a simpler way to improve the readability.

As described above, Figure 1C is an important visualization that enables users to assess the different modules that are driving functional performance and to uncover major biases. In this case, yes, the point we are illustrating with this plot is that the mitochondrial ribosome and ETC I modules explain the majority of the performance in the high-precision portion of the precision-recall curve in Figure 1B. However, in general, this type of plot is designed to highlight any bias that might appear in terms of functional modules and is one of the standard benchmarking outputs of the FLEX pipeline. We agree that our earlier version was unnecessarily complicated. To make this figure more easily interpretable, we have removed the bar plot that was previously to the right and have relabeled the X axis to make it more clear what is being plotted. We have

also modified the figure legend and the main text with an expanded description of the different plots produced by FLEX (see Minor point 1 for details).

3. In Fig. 1g, bottom panel, the curve of Wainberg *et al* does not start from $x=0$, probably due to visualization issue.

We thank the reviewer for noticing this detail; however, it is not a visualization issue. The Wainberg *et al.* approach produces a measure of statistical significance (FDR) for each gene pair analyzed. We are ranking pairs in descending order based on $(1-FDR)$ such that the most significant pairs appear at the top of the ranked list. There are a large number of gene pairs produced by this method with an $FDR=0$, which means there is no way to distinguish amongst this set. Due to these ties, this results in the first precision-recall point starting at a non-zero recall. This would be true of any method with ties among the highest ranking gene pairs. This information was originally mentioned in the Figure S4 (now Appendix Figure S4) legends, but we have now also added these details to the legend of Figure 1G: “The approach from Wainberg *et al.*⁷ bases gene pair similarity scores on FDR corrected p-values $(1 - \text{fdr})$ resulting in a ‘late start’ of the PR curve (many values at top are the same, 1.0).”

Reviewer #2

Rahman, Billmann and colleagues present a method (FLEX) for assessing the ability of co-essentiality, derived from CRISPR screens, to predict co-complex / co-pathway membership. Typically one might benchmark such approaches using a single list of true positives (e.g. all gene pairs that belong to the same complex) and a list of true negatives (e.g. all gene pairs that belong to different complexes). The authors show that such an aggregated evaluation set can mask important signals in the data - in particular for CRISPR screens they show that a couple of mitochondrial complexes are responsible for many of the true positives correctly identified by co-essentiality approaches. They develop, and advocate for, an approach to summarise the performance of co-essentiality across all individual modules. They also provide nice visualisations showing the contribution of each module to the predictive power of co-essentiality methods. Much of the manuscript is devoted to an analysis of which complexes are identified as essential in CRISPR screens performed by the Broad but not the Sanger. The authors provide a nice explanation (long protein half-lives and differing CRISPR screen lengths) for why some complexes are more essential in the Broad screens, but this analysis is largely orthogonal to the establishment of FLEX as a useful benchmark.

Overall FLEX seems like it might be a useful way of benchmarking approaches to identify functional interactions from co-essentiality, but there is very limited benchmarking performed in the paper. The analysis of which modules are essential in the Broad but not Sanger screens is interesting and the newly performed time-course CRISPR screens in HAP1s support the authors' hypothesis that screen duration is the major factor. However, previous work has discussed the issue of the consequences of the different assay lengths between the two

institutes and shown an enrichment of mitochondrial genes identified as uniquely essential in the Broad study (Dempster et al, Nature Comms 2019 <https://doi.org/10.1038/s41467-019-13805-y>). It's not clear that the FLEX analysis adds to this discussion.)

We thank the reviewer for this comment. We agree that more examples of benchmarking should be included, and we have added several based on this and other reviewers' suggestions (see more details in our reply to Major point 2). Regarding the question of what our manuscript adds to the previous discussion in the Dempster *et al.* paper: mitochondrial genes are mentioned one time in that paper: "The Broad-exclusive enriched GO terms included classes related to mitochondrial and RNA processing gene categories and other gene categories previously characterized as late dependencies". The authors identify a mitochondrial enrichment among genes that drop out late in screens. There's no discussion of the covariance observed for mitochondria genes across screens, which is the major focus of our findings here-- specifically, the dominance of this covariation on dependency networks derived from these data. Importantly, this variation is not just an issue when one compares the Sanger vs. the Broad screens: it is dominant even within a single dataset (e.g. the Broad screens). There, the authors focus on assay length, which is one factor that can cause differences in the apparent mitochondrial genes' phenotypes. However, even for a collection of screens run the same length (e.g. within the Broad screens only), we argue that there is an interplay between the doubling rate of the cell line being screened and global protein stability in each cell line, which will introduce variation in the phenotypes measured for mitochondrial genes. We also note that there is no discussion of the biological basis of the late drop-out of mitochondrial genes in the Dempster *et al.* manuscript, whereas we propose that this relates to protein stability. Thus, we believe our manuscript highlights several important insights that are not covered by the Dempster *et al.* paper.

More details on the additional benchmarking we've completed and added to the manuscript are described below in our reply to Major Point 2.

Major Points

It is well established that methods to predict functional associations between genes can be biased by certain complexes. Liu et al (NAR 2008, <https://doi.org/10.1093/nar/gkn972>) show that signals from using co-expression to predict complex membership are driven by a small number of large complexes (ribosome, proteasome). Often authors have developed ad-hoc solutions to address issues with large complexes dominating the signal e.g. Drew et al (MSB 2016 <https://doi.org/10.15252/msb.20167490>) exclude complexes with more than 30 subunits from their analysis. This does not diminish the utility of FLEX, but these existing approaches merit some discussion.

We agree with the reviewer that previous work on other types of genome-wide datasets have noted a problematic bias due to large complexes dominating functional evaluations of gene-pair data. As a tool for systematic evaluation of possible bias, FLEX was created to detect such biases by visualizing the contribution of each functional standard subset (module) to the global performance metric. It further contrasts the size of each module and its performance. This is crucial for several datasets evaluated in the current manuscript, because biases only affect a

specific set of large complexes (e.g. DepMap CRISPR screens) or can even be pronounced as expected biological bias (e.g. using data by Olivieri *et al*, 2020).

We have now discussed this in more depth and cited the above-mentioned papers at the end of the second paragraph of the subsection ‘*Development of a pipeline for evaluation of CRISPR screen data*’: “Similar issues have been reported when evaluating other types of genomic datasets in a pairwise manner, particularly for large, coherent protein complexes (Drew *et al*; Liu *et al*; Myers *et al*.)”

Given that the stated purpose of the tool is benchmarking, very little benchmarking is actually performed in the manuscript. Different groups have established different pipelines for scoring CRISPR screens (e.g. the Broad's DepMap project uses CERES, while the Sanger's project uses BAGEL and CRISPRcleanR) but these are not benchmarked. Furthermore, there are many ways of assessing co-essentiality (e.g. Pearson's correlation, Spearman's correlation, cosine similarity) and these are not benchmarked.

We thank the reviewer for the comments, and we agree that our manuscript would be improved with more examples of how FLEX can be used to benchmark various aspects of CRISPR screen interpretation. We previously included a benchmarking for two different versions of Broad DepMap co-dependency networks (original Figure S2) and another benchmarking for alternative methods for deriving co-dependency networks from the Broad DepMap (original Figure S6). To better emphasize those benchmarking efforts, we elevated the visibility of previous Figure S6 to Figure EV2.

To further address this comment, we performed three additional lines of benchmarking analysis with FLEX:

1. We benchmarked the effectiveness of different similarity measures (PCC, Spearman correlation, cosine similarity, and dot product similarity) for measuring co-dependency networks using FLEX (Figure EV1). Interestingly, this analysis suggests that Pearson and Spearman correlation are relatively similar to each other in their performance while cosine or dot product metrics clearly perform less well (Figure EV1).
2. We used FLEX to evaluate the effect of the number of screens included in the dataset on the performance of co-dependency networks by subsampling from Broad DepMap screens (Appendix Figure S5). More details on the results of this analysis are included in our response to Reviewer #1 (Major point 1).
3. We benchmarked previously published genome-wide RNAi screens against CRISPR (Broad DepMap) screens (Appendix Figure S10). Unsurprisingly, this revealed the superior performance of CRISPR screens in comparison to either of these RNAi screen datasets. A detailed result of the analysis is included in the response to the comment of Reviewer #1 (Major point 2).

We agree that these additional examples of benchmarking are useful for demonstrating the utility of FLEX.

To incorporate the benchmarking of similarity measures, we added the following text in the first paragraph of the subsection 'Example applications of FLEX to benchmark CRISPR screen data/analysis methods': "Second, we used FLEX to benchmark a variety of similarity metrics in their ability to construct co-essentiality networks that capture known functional relationships from the DepMap dataset. Specifically, we evaluated four different similarity measures for gene pairs: cosine similarity, inner (dot) product, Pearson correlation, and Spearman correlation. We found that Pearson correlation (PCC) and Spearman correlation provide comparable performance and that they clearly outperformed cosine and dot product similarity measures on the DepMap dataset (Figure EV1) (PCC is implemented as the default similarity measure in FLEX)."

To demonstrate the effect of screen sizes on function performance, we included the following as the second paragraph of the subsection 'Example applications of FLEX to benchmark CRISPR screen data and analysis methods': "In a fourth application example, we applied FLEX to explore the extent to which the ability to derive co-essentiality networks from a CRISPR screen dataset depends on the number of screens. Specifically, we subsampled different numbers of screens from the DepMap data, measured co-essentiality networks on the resulting datasets of varying size, and evaluated these scores for functional information using FLEX. Our analysis showed that the amount of functional information captured increases with the number of screens included as expected, but that this saturates relatively quickly (Appendix Figure S5). For example, FLEX analysis indicates that there is little measurable difference between the quantity of functional information captured by only 300 screens as compared to the complete collection of 563 in the 2019Q2 release of the DepMap data (Appendix Figure S5). Even a set of as few as 100 randomly sampled screens performs similarly to the complete set of 563. Our FLEX analysis also indicated that with 15 or fewer screens, the ability of co-essentiality scores to accurately capture functional information drops dramatically (Appendix Figure S5), suggesting this is a practical limit on the minimum number of screens required for generating co-essentiality maps."

Finally we added the results of systematic comparison of RNAi screens and CRISPR screens in the second paragraph of the subsection 'Exploring the basis of strong ETC-related co-essentiality relationships in CRISPR screens': "In our own FLEX-based analysis of RNAi screens (McFarland *et al*), we observed a similar, albeit weaker, enrichment for mitochondrial ribosome-related gene pairs (Appendix Figure S10), although unlike CRISPR screens, co-essentiality scores from RNAi screens also exhibited dominant enrichment for cytoplasmic ribosome gene pairs (Appendix Figure S10)."

Minor Points

Figure 2h - the fast slow arrows are extremely confusing. I initially thought they indicated that ETC/mtRibo were fast drop outs while spliceosome/proteasome were slow drop outs. The differing y-axes also make the chart harder to read.

We thank the reviewer for pointing this out and agree we could have made this plot more clear. Figure 2H plots a normalized estimate of the derivative of the log fold-change (LFC) for each drop-out profile. We've revised this panel by adding an inset plot to the left that clarifies what we

mean by “Dropout speed”. Specifically, we plot dropout speeds right above two hypothetical LFC profiles to provide a clear example of how dropout speed corresponds to the observed LFC values. We also updated the corresponding figure legend.

Figure update: Figure 2H, added left panel. Updated legend for 2H: “H, Dropout speed for ETC-related and other selected essential complexes. Dropout speed is a normalized estimate of the derivative of an LFC profile (across time) for each guide (see Methods). A positive dropout speed indicates faster relative dropout, while a negative dropout speed indicates slower dropout (see left panel for hypothetical LFC profile examples and their corresponding dropout speeds). The average dropout speed across all genes in each of the indicated complexes is plotted as a function of screen sampling time (right). tSNE embedding groups CORUM complexes with similar dropout speed (see Methods). The six selected complexes on the right are indicated in the tSNE plot (large colored dots) and sub-complexes are labeled with matching colors (bottom).”

Figure S4 - the different x-axes used for the module-level summaries make these charts very hard to compare. I thought PCC was performing at a similar level to Kim et al, although this is not the case

We apologize for the confusion. Figure S4 (now Appendix Figure S4) is designed to demonstrate the effect of mitochondrial bias removal on the performance of individual methods, not as a comparative benchmarking of those methods. The goal of Figure S5 (now EV2) is to provide a head-to-head comparison for those methods, and they are plotted on the same axes. We have modified the figure legend of now Appendix Figure S4 and Figure EV2 to reflect this and also added explanatory legends in Figure 1. The titles in the legends of these figures have been modified to better reflect their purposes (Appendix Figure S4 and EV2, respectively): “Exploration of mitochondrial bias of different DepMap post-processing approaches.” and “Direct comparison of alternative DepMap post-processing approaches.”

Reviewer #3

FLEX is a pipeline that is used to integrate, among others, publicly available reference datasets to evaluate the integrity of large-scale loss-of-function screening experiments. By using precision recall statistics, CRISPR screening dataset can be interrogated for known functional relationships between gene co-dependency gene pairs and pathways. The authors used FLEX to assess gene-level precision recall performance on commonly published datasets. Analyzing DepMap data based on a CORUM standard, they found that the genes part of the ETC I, V and the 55S mt ribosome complexes are largely dominating the co-essentiality dependency networks. To account for the large effect of certain pathways in determining gene-level PR, the authors developed modular PR to normalize highly co-dependent gene sets into one score. To explain this difference in ETC dependency, the authors hypothesized that the screen length between Broad and Sanger DepMap protocols are different, suggesting that the difference in

screen sampling times could contribute to the differences in essentiality. Notably, they have pointed out that the ETC I and V have the highest protein stability, while the 55S was comparatively close to the median half-life. To test this observation, the authors employed a CRISPR/Cas9 screen in HAP1 cells and sampled every 3-4 days. The authors identified several early and late dropout pathways and genes, highlighting the dependency of ETC-related genes in early time points. The authors suggested that the strength of ETC-related fitness is able to accurately predict the time of the screen. At the current stage, it requires a major revision. Specific comments are:

1. Overall, the manuscript presents a method to use a novel AUPRC method to identify differences in co-essential genes between studies. This represents a technical advance to anyone who wants to perform essentiality CRISPR-KO or other KO screens. However, it remains hypothetical that the authors claiming the differences in dropout rate for the screens could be explained by the protein half-life. In particular, the strength of ETC-related phenotype is proposed as a metric for determining the true length of a screen before sampling. While this point could be greatly beneficial, it is uncertain whether this can be true for Broad biological contexts. It is also unclear how useful this metric would be compared to any of the other essential genes or pathways. While the technical challenge has been clearly described, the reviewer is not convinced of how this finding would translate to promoting a clear biological advance.

(Note: we have moved a portion of comment (3) up as it relates to the reviewer's point (1) above)

3. The reviewer would like the authors to comment more on how monitoring ETC-related phenotypes can be applied to identify the difference in timing for screens between different cell types, and how tracking the dropout rate of ETC-related essential genes are different from doing so with other essential genes or complexes.

We briefly reiterate the key elements of our hypothesis here and then further address the reviewer's questions below. Our claim that the strength of ETC-related genes' phenotypes are a good indicator of screen timing is based on the data presented in Appendix Figure S6 and Figure 3C. Figures S8A-C demonstrate that the LFC (log fold change) values for mitochondrial ribosome and ETC I genes continue to grow more negative throughout the entire 18-day screen in HAP1 cells. In contrast, for other essential protein complexes (e.g. spliceosome, cytoplasmic ribosome, proteasome), the LFC values reach a minimum LFC very quickly (between 5-10 days), such that there is little variation observed in the LFCs for those complexes after 10 days. If differences in screen timing or growth rates of cell lines are limited to +/- ~50% of the "typical screen time" in a given context, one would expect then that only protein complexes like the mito. ribosome and ETC I genes would show different phenotypes related to such differences in timing - the other essential complexes drop out so quickly that there is negligible variation in phenotypes for the vast majority of screens, even with some variation in effective sampling time. This idea that the ETC-related genes can effectively serve as a clock for a CRISPR screen is further supported by the data presented in Figure 3C. We ranked our HAP1 screen individual timepoint LFC measurements for ETC-related genes amongst the entire DepMap collection

based solely on the strength of the observed ETC-related gene phenotypes. This quantity correlates well with the time at which the sample was taken (Spearman $r=0.61$, $p = 0.0007$, Figure 3C). When we repeated the same analysis with the other protein complexes, none of the correlations for the other essential complexes (spliceosome, cytoplasmic ribosome, or proteasome) were significant ($p > 0.05$ for all 3). To make this logic more clear, we have now included the following text in the second paragraph of 'Discussion' section:

"Why are ETC-related genes unique in this regard? If differences in the effective sampling timing or growth rates of cell lines are limited to +/- ~50% of the typical sampling time in a given collection of screens, one would expect that only protein complexes like the mitochondrial ribosome and ETC I genes, whose fitness effect size is still increasing even late in screens would show different phenotypes related to such differences in timing. Other essential complexes drop out rapidly enough that there is negligible variation in phenotypes for the vast majority of screens regardless of small variation in effective sampling time or other factors."

We agree with the reviewer that our proposal about the connection between protein stability and the strength of ETC-related gene phenotypes is "hypothetical" in some sense, but we have presented multiple lines of evidence that support this hypothesis. We agree that more definitive experiments could be done to further test this hypothesis and have edited the text of the Discussion to reflect this:

"We note that while multiple lines of evidence support our hypothesis about the effect of protein stability on ETC-related genes' phenotypes in CRISPR screens, more definitive experiments could be done to further test this hypothesis. For example, one could specifically quantify the dynamics of wild-type protein abundance in a population of cells expressing guides targeting ETC-related genes. Also, we note that there are likely additional non-genetic factors (e.g. beyond sampling time, growth rate and protein stability) that could similarly modulate the apparent phenotypes measured in CRISPR screens."

It's worth noting, however, that one key aspect of our paper is not hypothetical-- the fact that mitochondrial-related genes dominate evaluations of co-essentiality scores and relative comparisons of different methods for generating them. We clearly demonstrate the impact that these genes have on these evaluations, an effect that is important for our field to be aware of, regardless of the source of this bias.

Regarding this comment, "the reviewer is not convinced of how this finding would translate to promoting a clear biological advance": our results suggest that there is substantial variation in the measured phenotype for ETC-related genes in CRISPR screens that is not due to differences in genetic dependency. This finding is important for anyone applying CRISPR screens to identify context-specific genetic dependencies. While we discovered the effect based on the DepMap dataset, we expect that these factors also influence small-scale CRISPR screens (currently being widely applied across our community) and will result in ETC-related genes being identified as "hits" when they are actually not true differential genetic dependencies. Differential phenotypes for these genes should be interpreted with caution. Thus, we believe our findings do have broad relevance to our field. To make this point explicit, we have added this to the second paragraph of 'Discussion' section:

"While this effect is readily discoverable in the DepMap dataset, phenotypes for these ETC-related genes should be interpreted with caution in other CRISPR screen contexts as well,

especially if one is interested in scoring differential phenotypes (e.g. cell line-specific dependencies, genetic- or chemical-genetic interactions).”

2. The manuscript is written in a blended way, combining both a computational method with some related discoveries/hypotheses. It is confusing as the title only states one of the technical challenges in CRISPR screening analysis. There is a lack of clarification of challenges in the introduction and their approach to address these challenges. There is also a lack of summary on how their method and their discoveries can change the field. Much efforts are needed to make the whole manuscript more consistent, clearer, and less confusing.

We thank the reviewer for this comment. In our original version, we attempted to cover both the FLEX pipeline and our mitochondria-related gene finding in a very succinct manner. We understand why this lacked clarity and have modified several aspects of the manuscript based on the reviewer’s comment. Specifically:

- We changed the title to “A method for benchmarking genetic screens reveals a predominant mitochondrial bias” to better reflect both the introduction of the FLEX pipeline as well as our specific finding about mitochondria-related genes.
- We expanded both the abstract and introduction to include a focus on both the method and the mitochondrial gene finding.
- We expanded our description of the actual functionality of FLEX in the early part of the Results section to make it more clear what problems it addresses.
- We added a section in the Results that highlights other examples of benchmarking to demonstrate the types of questions our FLEX pipeline can help answer.
- We added subsection headings to help with clarity of organization of the manuscript.
- We expanded the Discussion section to include the following:
 - A clear statement that we think differential phenotypes for ETC-related genes should be interpreted with caution in all CRISPR screens, emphasizing the broad relevance of our finding beyond interpretation of the DepMap dataset.
 - Several sentences clarifying the scope of the FLEX pipeline and other potential applications, indicating that it can be applied to evaluate gene-pair relationship data of any type using a variety of different standards.
 - Statements regarding the primary limitations of the FLEX pipeline.

Due to the extensive nature of these edits, we haven’t copied them here. However, we included a version of our manuscript in our submission files that highlights all of our changes. We agree that these changes have improved the clarity of our manuscript.

(Note: this is the remainder of this reviewer’s comment (3)--the first part was moved up)

3. (continued) Do the cells applied with different drugs or other treatments have similar essentiality scores for these genes? The reviewer would like the FLEX analysis of a published screen with cell and/or animal models with drug treatments such as those found in chemotherapy studies to compare the AUPRC.

We thank the reviewer for this suggestion. To explore this, we applied FLEX to 31 targeted chemical genetic screens from Olivieri *et al*, 2020 (Appendix Figure S6). These screens were performed on RPE-1 cells and include a collection of drugs targeting DNA-damage related pathways. This evaluation shows that a co-dependency network derived from these chemical genetic screen profiles does predict functional relationships (Appendix Figure S6A), although with substantially lower performance than the complete DepMap dataset (Appendix Figure S5A) or even an equivalent number of screens sampled from the DepMap (Appendix Figure S5F). Interestingly, the FLEX contribution diversity analysis indicates that the vast majority of true positive relationships derived from these data are from protein complexes related to DNA damage repair (e.g. Fanconi anemia complex, DNA ligase IV–XRCC4–XLF complex, DNA synthesome complex) (Appendix Figure S6B), which reflects the focus of the chemical genetic screens. We did not see a strong representation of ETC-related complexes in this co-dependency network, suggesting the factors driving the variation in ETC-related genes' phenotypes are less prominent in this context. In general, this analysis demonstrates the utility of FLEX for interpretation of other CRISPR screens beyond the DepMap dataset. We have added a summary of this additional application of FLEX to the manuscript in the last paragraph of subsection 'Example applications of FLEX to benchmark CRISPR screen data and analysis methods' and it is also included below:

“As a final example FLEX application, we explored the question of how the identity of genetic screens affects the type of functional information captured in co-essentiality scores. Specifically, we applied FLEX to analyze the co-essentiality scores derived from 31 genome-wide CRISPR-Cas9 screens against 27 DNA-damaging agents (Olivieri *et al*). Interestingly, FLEX contribution diversity analysis showed a strong dominance of protein complexes related to DNA damage repair (e.g. Fanconi anemia complex, DNA ligase IV–XRCC4–XLF complex, DNA synthesome complex) among predicted functional relationships (Appendix Figure S6A, B). At the same time, ETC-related complexes were not strongly represented amongst these co-essentiality scores, suggesting that the factors driving the variation in ETC-related genes' phenotypes are less prominent in this context. This example more generally shows how the biological focus of the investigated set of screens, an experimental theme spanning various model organisms (Billmann *et al*; Jonikas *et al*), can be evaluated.”

Aside from protein degradation, as suggested by looking at the protein half-life in Figure 2b, there could be other biological reasons for functional differences in dropout speed including different levels of essentiality to the cell. For example, a gene that is required for mismatch repair may drop out much slower than a gene required for cell division checkpoints. Could the proteins of ETC genes be more tolerant of DNA mismatches? In an effort to isolate protein stability from the other biological effects, the reviewer suggests the authors to provide protein stability-corrected essentiality scores for the genes based on the 18,000 gene sgRNA library applied to the HAP1 cells.

We completely agree with the reviewer that in general, there could be many factors that influence the dropout speed of a gRNA targeting a particular gene including the specific function of that gene or the magnitude of the fitness effect caused by the loss of that gene's function.

However, in the case of the ETC-related genes, we do not think it is coincidental that these proteins exhibit the highest stability of any protein complex in the proteome. Our time-resolved screens in HAP1 cells also provide additional evidence for this hypothesis in that they demonstrate that the dynamics of the dropout of ETC-related genes are unique (consistent with their unique prolonged protein stability) and that these dynamics alone are enough to explain the range of phenotypes observed in the DepMap (Figure 3C).

In the case of this complex, we think it is unlikely that the reason for the slow dropout is related to tolerance of DNA mismatches. The guides in our gRNA library induce a variety of insertions and deletions, many of which will cause out-of-frame translation. The additional tolerance to DNA mismatches hypothesis would require a substantial number of in-frame indels induced by the set of guides targeting these genes, and furthermore, it would require that most of the ETC-related genes have a similar tolerance for mismatches.

We agree with the reviewer that it is important to acknowledge in the manuscript that while we present evidence for the protein stability hypothesis, more definitive experiments could be done to further test this hypothesis. We have edited the text of the Discussion to reflect this and have also mentioned the possibility that there are many other factors that contribute to the strength of CRISPR screen phenotypes.

Regarding providing stability-corrected essentiality scores, we are currently working on a method for correcting these effects and agree that this is an important direction. However, as the reviewer notes above, the current manuscript is already stretched in the sense that we are both introducing the FLEX method that enabled the discovery of the ETC-related bias as well as exploring the biological basis of this bias (both important in our opinion). Describing a method for normalizing this effect, along with proper analysis to demonstrate that it works, is beyond the scope of this manuscript.

We updated the text in the Discussion section in response to this comment: “We note that while multiple lines of evidence support our hypothesis about the effect of protein stability on ETC-related genes’ phenotypes in CRISPR screens, more definitive experiments could be done to further test this hypothesis. For example, one could specifically quantify the dynamics of wild-type protein abundance in a population of cells expressing guides targeting ETC-related genes. Also, we note that there are likely additional non-genetic factors (e.g. beyond sampling time, growth rate and protein stability) that could similarly modulate the apparent phenotypes measured in CRISPR screens.”

4. In Figure 2h, the authors highlight the differences in protein degradation speeds as they relate to dropout rate. The reviewer would like the authors to comment towards if there is a delayed fast or a delayed slow rate, and if 20 days is enough to show these curves. The reviewers would also like the authors to comment on why the differences in essential/non-essential gene separation in S6C seem to dissipate after 10 days while the ETC complex displays a dropout difference between the Sanger and Broad screens which are between 14 and 21 days?

We think there was some confusion about 2H, and we understand that our original version of this figure lacked clarity (also commented on by Reviewer #2). To clarify, Figure 2H plots the

dynamics of the dropout for different protein complexes, including the ETC-related genes, but does not plot any data associated with protein stability. Our goal in showing the details of these dynamics is to explain how the increased protein stability of ETC-related genes could result in variable phenotypes across a set of screens. As described above, we added an additional panel to Figure 2h and extra detail to the legend.

Figure update: Figure 2H, added left panel. Updated legend for 2H: “H, Dropout speed for ETC-related and other selected essential complexes. Dropout speed is a normalized estimate of the derivative of an LFC profile (across time) for each guide (see Methods). A positive dropout speed indicates faster relative dropout, while a negative dropout speed indicates slower dropout (see left panel for hypothetical LFC profile examples and their corresponding dropout speeds). The average dropout speed across all genes in each of the indicated complexes is plotted as a function of screen sampling time (right). tSNE embedding groups CORUM complexes with similar dropout speed (see Methods). The six selected complexes on the right are indicated in the tSNE plot (large colored dots) and sub-complexes are labeled with matching colors (bottom).”

Regarding the question about essential/non-essential gene separation in our HAP1 screen data at 10 days, we would like to note that both early essential (e.g. 26S proteasome) and late essential (e.g. 55S ribosome) genes tended to reach a ‘bottom’ around day 10 (see Fig. EV3C). To explain why additional ETC-related gene dropout occurs between day 14 and day 21 in the DepMap screen data, we provide two reasons (both likely contribute):

1. The definition of assay length is reported differently in our experiment (days past puromycin selection for gRNA containing cells) than in the DepMap protocol (days past gRNA library transfection). In our protocol, gRNA library transfection takes place roughly 4 days prior to successful puromycin selection. Therefore, our day 10 corresponds to day 14 in the DepMap protocol.
2. Different cell lines can have substantially different doubling rates. HAP1 cells proliferate rather quickly, and presumably, more quickly than several cell lines screened in the DepMap project. Since doubling rate is a main determinant of residual wildtype protein levels (as we describe in our manuscript), the ‘bottom’ observed in HAP1 around day 10 would be reached later in those slower cell lines.

Thank you for sending us your revised manuscript. We have now heard back from the three reviewers who were asked to evaluate your study. As you will see the reviewers are overall satisfied with the modifications made and think that the study is now suitable for publication.

Before we can formally accept your manuscript, we would ask you to address the following issues.

REFEREE REPORTS

Reviewer #1:

I appreciate the authors' efforts to answer my questions in the 1st round review. Most of the points have been addressed well. My remaining concern lies on Point 3, in which I asked for the examination of the viability profile upon KO of other stable proteins in addition to ETC complex subunits. To my understanding, the results from the authors' analysis suggest weak or even no correlation between other stable proteins and the ETC complex subunits. While I do not exclude the possibility that protein stability contributes to the high similarity of ETC-related co-essentiality, I think other factors, such as the redox potential of the medium in cell culture, may also explain the observed similarity in an alternative way. I'd suggest the authors to include the analysis on Point 3 and to mention it in the 2nd discussion paragraph "Why are ETC-related genes unique in this

regard?"

Reviewer #2:

Billman and colleagues have addressed my concerns. The manuscript has been significantly improved by the addition new benchmarking analyses - comparing different correlation metrics and comparing co-essentiality derived from varying numbers of cell lines.

Reviewer #3:

Since the original review, the reviewers have addressed deficiencies in explanation for the scope and applications of this technology, and have more clearly defined FLEX's benchmarking capabilities. The claims that highly stable protein complexes such as those involving ETC-related genes have been largely reduced, with the surviving conclusion being mitochondrial-related genes dominate evaluations of co-essentiality scores for DepMap gene sets. The reviewer thanks the authors for providing the experiment for looking at the DNA damage-related screening study and identifying different pathway dependencies there. Overall, the reviewer is satisfied with the amount of work done to address the comments, though the conservativeness of the corrected conclusions may reduce the impact of the paper slightly.

Response to reviewers

Reviewer #1:

I appreciate the authors' efforts to answer my questions in the 1st round review. Most of the points have been addressed well. My remaining concern lies on Point 3, in which I asked for the examination of the viability profile upon KO of other stable proteins in addition to ETC complex subunits. To my understanding, the results from the authors' analysis suggest weak or even no correlation between other stable proteins and the ETC complex subunits. While I do not exclude the possibility that protein stability contributes to the high similarity of ETC-related co-essentiality, I think other factors, such as the redox potential of the medium in cell culture, may also explain the observed similarity in an alternative way. I'd suggest the authors to include the analysis on Point 3 and to mention it in the 2nd discussion paragraph "Why are ETC-related genes unique in this regard?"

We thank the reviewer again for placing our efforts into a larger context and agree that other experimental factors in CRISPR screens such as the redox potential of the cell culture media likely do impact a cell's dependency on the electron transport chain (ETC). In fact, a meta-analysis of the DepMap data with a focus on media conditions, which we also cite in our manuscript, found differential gene effects including the gene *ASNS* (Lagziel *et al*). That study also built predictive models for cell line-specific dependence on genes in one-carbon metabolism. However, this study did not show a major differential effect for either ETC complexes I and V or the 55S ribosome. Moreover, this work predicted experimental factors explaining screen-to-screen differences and neither the ETC nor protein stability was mentioned.

Nonetheless, we agree with the reviewer that other factors may contribute to the observed ETC-related gene co-essentiality. We have updated the suggested paragraph of the discussion to include this:

"Also, we note that beyond sampling time, growth rate and protein stability, there are likely additional non-genetic factors, such as the redox potential of the media as explored by Lagziel and colleagues (Lagziel *et al*), that could similarly modulate the apparent phenotypes measured in CRISPR screens."

We have not added any supplemental analysis related to our earlier response to Point 3. It is important to note that the ETC complexes are not only the highest in terms of protein stability of any complex, they are the highest by a considerable margin. The median protein half-life of the ETC V complex across B cells, monocytes and hepatocytes is about 11 days while the median protein half-life of the next most stable complex is 6.5 days, and the overall median half-life of any complex is 3 days as reported in the original study of protein stability (Mathieson *et al*). Given this considerably higher protein stability, the significance of the absence of dropout profile similarity of the next most stable complexes is unclear to us. We feel that citing this negative result with unclear implications will only create confusion.

Reviewer #2:

Billman and colleagues have addressed my concerns. The manuscript has been significantly improved by the addition new benchmarking analyses - comparing different correlation metrics and comparing co-essentiality derived from varying numbers of cell lines.

We thank the reviewer again for the constructive suggestions on our manuscript.

Reviewer #3:

Since the original review, the reviewers have addressed deficiencies in explanation for the scope and applications of this technology, and have more clearly defined FLEX's benchmarking capabilities. The claims that highly stable protein complexes such as those involving ETC-related genes have been largely reduced, with the surviving conclusion being mitochondrial-related genes dominate evaluations of co-essentiality scores for DepMap gene sets. The reviewer thanks the authors for providing the experiment for looking at the DNA damage-related screening study and identifying different pathway dependencies there. Overall, the reviewer is satisfied with the amount of work done to address the comments, though the conservativeness of the corrected conclusions may reduce the impact of the paper slightly.

We thank the reviewer again for the constructive suggestions on our manuscript.

Thank you again for sending us your revised manuscript. We are now satisfied with the modifications made and I am pleased to inform you that your paper has been accepted for publication.

Corresponding Author Name: Maximilian Billmann, Chad L. Myers

Manuscript Number: MSB-20-10013